# Mapping membrane biophysical nano-environments

Luca Panconi [1,2,3,9], Jonas Euchner [3,4,5,9], Stanimir A. Tashev[3,4,5], Maria Makarova[3,6,7], Dirk-Peter Herten [3,4,5], Dylan M. Owen [1,3,8] & Daniel J. Nieves [1,3] ✉

The mammalian plasma membrane is known to contain domains with varying lipid composition and biophysical properties. However, studying these membrane lipid domains presents challenges due to their predicted morphological similarity to the bulk membrane and their scale being below the classical resolution limit of optical microscopy. To address this, we combine the solvatochromic probe di-4-ANEPPDHQ, which reports on its biophysical environment through changes in its fluorescence emission, with spectrally resolved single-molecule localisation microscopy. The resulting data comprises nanometre-precision localisation coordinates and a generalised polarisation value related to the probe's environment – a marked point pattern. We introduce quantification algorithms based on topological data analysis (PLASMA) to detect and map nano-domains in this marked data, demonstrating their effectiveness in both artificial membranes and live cells. By leveraging environmentally sensitive fluorophores, multi-modal single molecule localisation microscopy, and advanced analysis methods, we achieve nanometre scale mapping of membrane properties and assess changes in response to external perturbation with methyl-β-cyclodextrin. This integrated methodology represents an integrated toolset for investigating marked point pattern data at nanometre spatial scales.

The mammalian plasma membrane is crucial for cellular function. It is involved in transport, mechanotransduction, and signalling, with its biophysical properties playing a key role in regulating these processes. One such property is membrane lipid order, where it is hypothesised that the plasma membrane can partition into disordered and ordered phase regions[1,2]. Ordered regions are enriched in saturated lipids and sterols, as well as proteins which have high affinity for that environment, such as those with longer transmembrane regions or saturated post-translational acylation. Ordered regions also display tighter lipid packing, thereby limiting the penetration of polar water molecules into the bilayer core[2]. Conversely, disordered regions are enriched in unsaturated-tail phospholipids and display looser lipid packing, with consequent increased water infiltration. While readily observed in artificial membranes, there is ongoing debate about the

[1]Department of Immunology and Immunotherapy, School of Infection, Inflammation and Immunology, College of Medicine and Health, University of Birmingham, Birmingham, UK. [2]School of Physics and Astronomy, College of Engineering and Physical Sciences, University of Birmingham, Birmingham, UK. [3]Centre of Membrane Proteins and Receptors, University of Birmingham, Birmingham, UK. [4]Department of Cardiovascular Sciences, School of Medical Sciences, College of Medicine and Health, University of Birmingham, Birmingham, UK. [5]School of Chemistry, College of Engineering and Physical Sciences, University of Birmingham, Birmingham, UK. [6]School of Biosciences, College of Life and Environmental Science, University of Birmingham, Birmingham, UK. [7]Department of Metabolism and Systems Science, School of Medical Sciences, College of Medicine and Health, University of Birmingham, Birmingham, UK. [8]School of Mathematics, College of Engineering and Physical Sciences, University of Birmingham, Birmingham, UK. [9]These authors contributed equally: Luca Panconi, Jonas Euchner. ✉e-mail: d.j.nieves@bham.ac.uk

existence and nature of ordered lipid microdomains in complex cell membranes[1].

Differences in lipid packing can be detected by solvatochromic probes[3]. Water, a polar molecule, penetrates deeper into the liquid-disordered phase of the membrane and increases the overall polarity of the probe's environment[4]. This difference in polarity yields a change in the emission spectra of the probe[5]. Nile Red, Laurdan and di-4-ANEPPDHQ have previously been employed in this context, with di-4-ANEPPDHQ showing a 60 nm shift toward shorter emission wavelengths within ordered membrane regions (Fig. 1a). Such probes have been used with confocal microscopy, 2-photon imaging, TIRF and other modalities to map membrane lipid order[6]. However, membrane domains are thought to exist below the resolution limit of conventional microscopy[7,8].

Super-resolution methods are a family of techniques that bypass the classical resolution limit. Past work has therefore sought to utilise super-resolution and single molecule methods to try to measure lipid domains at the nanoscale[9]. For instance, pioneering work by Eggeling et al. utilised the trapping of different lipid based probes in nanoscale domains using stimulated emission depletion (STED) microscopy[10] and extensions to the technique (i.e., STED-FCS[11]). Similarly, approaches employing single particle tracking have also been used to determine the dimensions of lipid-based labels within domains based on their confinement behaviour[12,13]. Single molecule localisation microscopy (SMLM) on the other hand is a promising method for membrane measurements at the nanoscale due to its comparatively higher spatial resolution and has been combined successfully with environmentally sensitive membrane probes[14–16]. For example, Nile Red has been imaged using points accumulation in nanoscale topography (PAINT) and spectrally resolved SMLM on artificial membranes and in other hydrophobic environments[14,17].

Fundamentally, SMLM does not produce images. Instead, it generates a list of the x,y coordinates of the detected molecules—a point pattern[18]. The resulting data from combining spectrally-resolved SMLM with environmentally sensitive probes takes the form of a marked point pattern, i.e., each x,y-coordinate is associated with one (or more) values relating to the spectral properties of the probe. Here, the mark is the calculated generalised polarisation (GP) value, essentially a normalised intensity ratio between two channels (Fig. 1b). GPs can range from -1 to +1, with high GP values deriving from ordered regions and low GP values from disordered regions.

Here, we use di-4-ANEPPDHQ to resolve the nanoscale membrane order of live cells and analyse it using our umbrella software package, PLASMA (Point Label Analysis of Super-resolved Mark Attributes), containing three analysis algorithms specifically designed for visualisation and interpretation of marked point patterns derived from SMLM. We use PLASMA to detect and then map membrane lipid nanodomains in live cells, using SMLM.

With the development of improved fluorescent probes which encode properties about the samples, for example pH[19] or membrane tension[20], there is great potential for generating information-rich marked point patterns that go beyond the standard structural information usually obtained via SMLM. Therefore, approaches to analyse marked point pattern data such as PLASMA will be of increasing interest.

## Results

### Di-4-ANEPPDHQ based ratiometric PAINT enables generation of marked point data

To generate marked point patterns from SMLM data, the probe must encode both its spatial position and some information about the property of its environment within its fluorescence emission. Solvatochromic probes (and their modified versions) have shown great potential for acquiring such data[3,17,21]. Di-4-ANEPPDHQ is an attractive option, as it yields a large spectral blue shift in its emission, reflecting the decreased polarity and lower hydration of the membrane (Fig. 1a),

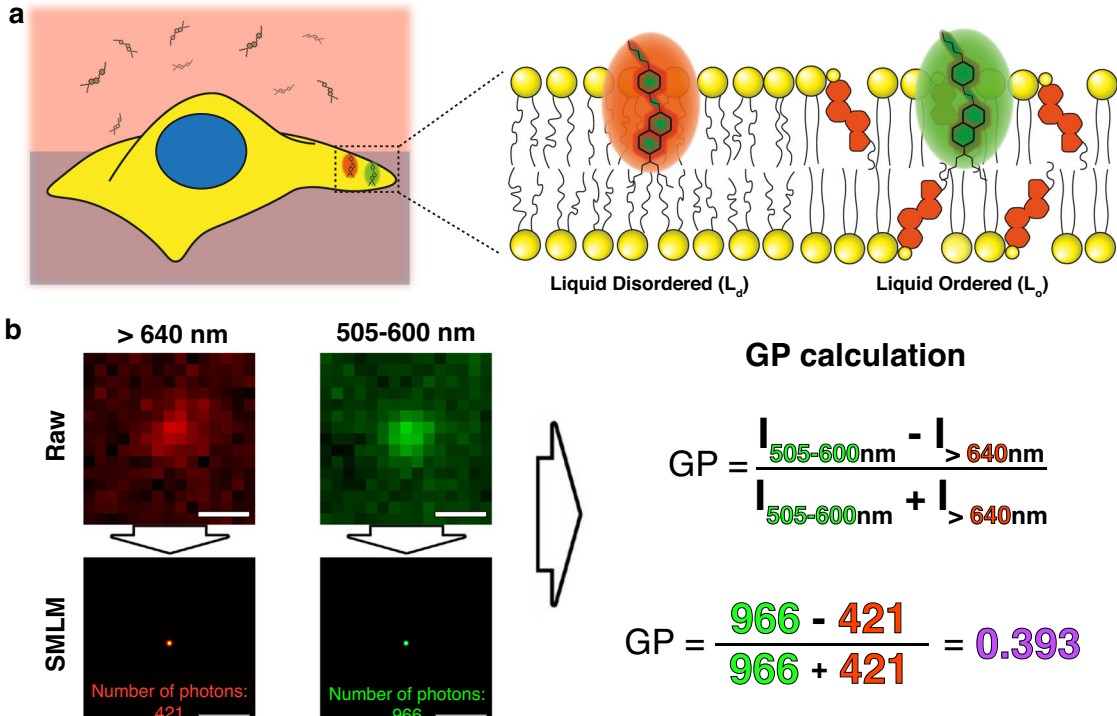

**Fig. 1 | Principle and generation of marked di-4-ANEPPDHQ PAINT data.**
**a** Schematic of the di-4-ANEPPDHQ PAINT principle. An excess of di-4-ANEPPDHQ probes rapidly diffuse in solution and insert into the membrane upon contact (left). The emission spectra of the individual probes are determined by the polarity of the lipid environment (right). **b** Exemplar derivation of GP values from experimental ratiometric di-4-ANEPPDHQ PAINT data. Raw PSFs are localised to x, y coordinates. From the photon count associated with each coordinate in each channel, GP values are calculated. Scale bars 500 nm.

which would be expected for single emitters, in line with literature on bulk measurments[5]. Further, di-4-ANNEPDHQ has been shown to be unaffected by proteins inserted within the membrane, thus its spectral behaviour will depend on lipid packing[22]. To be sensitive to the spectral shifts of single di-4-ANEPPDHQ probes the emission was split between two channels (Ch1: 505–600 nm and Ch2:> 640 nm; Fig. 1b and Supplementary Movie 1). At concentrations between 20–80 nM, with an average of $1825 \pm 688$ localisations were collected per $\mu m^2$ (1 localisation/584 $nm^2$, i.e., approximately one localisation every $23 \times 23$ nm region, after filtering) within the membrane, and negligible background binding events (total acquisition time: 8 mins). Di-4-ANEPPDHQ does bind to the cover glass, however, the density of the points that survive our filtering parameters are much further reduced when compared to specific binding to membranes (Supplementary Fig. 1). For cells, we focused on the apical membrane as binding of the dye is only visible on this membrane in the short imaging time window. Thus, background binding events on the glass were largely absent in the final data, post-filtering. Furthermore, no endogenous signals were observed within the cell that could contribute to erroneous localisations, i.e., only low intensity autofluorescence signals were observed within the 505-600 nm channel, with no corresponding signals in the far-red channel ( > 640 nm; Supplementary Movie 1; Panels A and B). For true binding events within membranes environments 50–5000 photons per channel were collected for each localisation (Supplementary Movie 1; Panels C and D). Photon yields within each channel were then used to calculate the GP value at the emitter position (Fig. 1b), thus generating marked point pattern data.

## Detecting domains in marked point patterns and assigning statistical significance

An important goal of point pattern analysis is assessing the presence of underlying features present within the data, e.g., non-uniform spatial arrangements and clustering. In the context of the plasma membrane, the ability of analytical methods to detect domains within marked point pattern data from SMLM experiments is critical. For example, the spatial distribution of points may be random, however, the properties, or marks, associated with the points may have underlying organisation. Therefore, we developed an analysis, P-Check, to determine if a point pattern expresses mark heterogeneity, i.e., *whether domains are present within the data*. After binarizing the marks into high and low GP categories (Fig. 2a), P-Check determines the proportion of neighbouring points within a defined search radius ($r = 50$ nm) that share the same mark (here, membrane lipid order) as a point of interest (Fig. 2b, c) and calculates the probability ($P_0$) that neighbouring points belong to the same group (see "Methods") (Fig. 2d). To obtain significance we randomly permutated the marks, keeping the spatial coordinates fixed. $N = 100$ permutations are performed, the value $P_i$ is calculated for each, and the ranking of $P_0$ in the 100 $P_i$ values determined. $P_0$ is maximised if points belonging to the same mark category (high order and low order) form distinct domains (Fig. 2a–d).

To demonstrate the principle of P-Check, we simulated 1000 different point patterns from which each mark was derived from one of two possible Normal distributions representing the range of GP values obtained from ordered and disordered regions (see "Methods"). Each pattern was discretised using Gaussian mixture modelling to recover the original point of intersection of the two Normal distributions[23], and this value was used as a threshold to separate points into two categories, i.e. ordered (high GP) and disordered (low GP) (Supplementary Fig. 2).

Data was simulated with increasing overlap of the two Normal distributions of the mark, and $P_0$ was calculated. This revealed that $P_0$ decreases as the mark overlap of the two distributions increases (Fig. 2e). This is because of the difficulty to distinguish differences in high and low order points due to the GP mark categorisation increasing with overlap of the distributions (Supplementary Fig. 3). For

low overlap, a large $P_0$ value is returned which surpasses all permutations (Fig. 2f), whereas for high overlap, the $P_0$ value returned is low with no significant evidence of separation found (Fig. 2g). Overall, P-Check can determine, with statistical significance, whether there are domains (here, GP) in the data. In general, achieving statistical significance can be aided if the point coordinates are spatially clustered and by having low overlap between the distribution of marks in each category.

## Visualising membrane domains - Membrane Order Mapping (MOM)

When detecting the GP from single polarity-sensitive dye molecules, the photon counts in each channel can be noisy[24]. In conventional fluorescence imaging, GP values are spatially-averaged over potentially hundreds of dyes, into the pixels and individual variations in GP values will be masked. Given di-4-ANEPPDHQ PAINT data contains the photon yield for each emitter within the two spectral channels, spatial averaging can serve as an avenue to limit these noise contributions that may arise from single emitters. Here we introduce a method of spatially averaging photon counts in each channel, but maintaining sub-diffraction information, and using the resulting histograms to derive a nanoscale membrane order map (MOM, see "Methods") (Fig. 3). MOM allows for visualisation of marked point patterns. This can be used for ascertaining preliminary qualitative results, and acts as a precursor for further statistical analysis, given that existing image analyses can now be applied to this data (Supplementary Fig. 4).

## Quantitative mapping of domains in marked point patterns

Having demonstrated domains can be detected by P-Check and visualised with MOM, the next goal was to identify and segment them. To this end, we developed an approach, termed Justification of Separation by Employed Persistent Homology (JOSEPH). Briefly, JOSEPH identifies groups of points which are both spatially correlated and have marks lying within a range of the group's mean[25], i.e., domains. Topological analysis methods are advantageous here, as they do not rely on specific data geometries, and will instead identify clusters irrespective of cluster shape[26,27]. Further, persistent homology has been used to probe qualitative features of data sets, and then as a precursor to cluster analysis[28]. In JOSEPH, we employ persistent homology to construct clusters in which points co-localise spatially and share little variation among their marked values.

The steps in JOSEPH analysis are shown in Fig. 4. First, the point pattern was pre-processed to construct a neighbourhood around each point with the search radius, $r$. Each point is then assigned a neighbourhood comprised of all other points within the search radius (Fig. 4a). The similarity, i.e., the difference between the mark of each point and the mean of all the points in the neighbourhood, was calculated and normalised to give a value between 0–1 (Fig. 4b, c). We then ordered the points from highest to lowest similarity and connect points with neighbours that have the highest similarity. This iterative process is terminated once a defined threshold, the deviance ($\tau$), is reached. Optimal values for both, $\tau$ and $r$, are automatically estimated from the data (see Supplementary Information). In each iteration of the algorithm, we link each point to neighbouring points that have higher similarity scores and whose central node (or 'root') has a mark value no more than $\tau$ higher than the point's own. If this linking action connects two previously separate clusters, the cluster with the lower similarity root is absorbed into the other. The clusters formed through this process provide a close approximation of the underlying domains within the data (Fig. 4d, e).

We evaluated the performance of JOSEPH using simulated data sets of varying cluster geometries, densities, and mark distributions as ground truth (see "Methods"). For each simulation we recorded the domains found by JOSEPH and calculated the intersection over union (IoU) score[29]. Briefly, the IoU quantifies how well identified domains

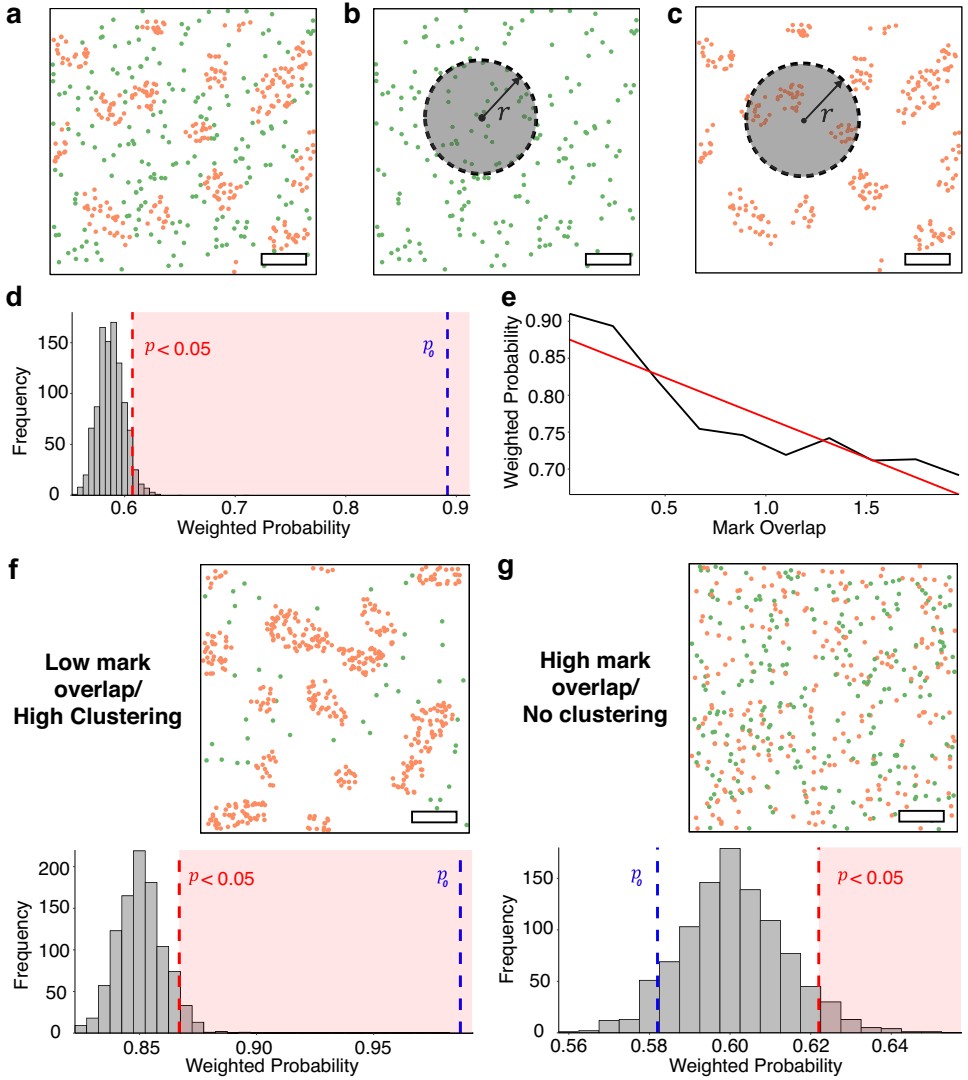

**Fig. 2 | Using P-check to detect domains in simulated marked point patterns.**
**a** A discrete point pattern can be segmented into subset point patterns for each category – low order points (green) and high order points (orange); **b**–**c**. Frequencies are calculated to determine the total number of neighbours for each point belonging to each of the two categories. **d** These values are summed over all points and the ratio gives the weighted probability, which can be compared to permutations of the same data set. **e** Mean probability returned from simulated data sets with varying degree of overlap of the GP values from each category.

**f** Example simulated data set with low overlap and pronounced clustering which shows statistically significant segregation of the mark (GP) values. **g** Example simulated data set with high mark overlap over a spatially random distribution, leading to no statistically significant separations of the marks (e.g., GPs). P-Check uses a one-sided permutation test with Bonferroni correction to adjust for multiple comparisons. The estimated $p$-values for Fig. 2d, f and g are 0.001, 0.001 and 0.921, respectively. P-Check was tested on 1000 simulated data sets and was repeated 3 times. Scale bars –500 nm.

overlap with the ground truth, with no overlap giving a value of 0 and for perfect overlap, a value of 1. We identified two important variables that impact JOSEPH's performance. Firstly, the overlap between the distributions of mark values (Fig. 4f), i.e., if the mark values within spatially clustered points are similar to those outside of the clusters, then domains will not be identified. Secondly, the proportion of points assigned to domains (Fig. 4g), i.e., if the density of points within the data, particularly within domains, is low then it becomes more difficult to identify domains.

### Demonstration on artificial membrane data
We first acquired data using established giant unilamellar vesicle (GUV) preparations for high and low membrane order (DPPC with 30 mol% cholesterol and pure DOPC, respectively), which were osmotically popped to produce a planar membrane patch (Fig. 5a, b). 20 nM di-4-ANEPPDHQ was then added to the sample after membrane deposition.

Di-4-ANEPPDHQ was detected upon insertion into the bilayer, allowing SMLM imaging in a PAINT-like modality. At each insertion event, a blink was detected by the camera in the two channels. Localisations (x,y coordinates) were determined, and a GP value for each event calculated from the derived photon count in each of the two channels, as in (Fig. 1b).

Examples of two MOMs from each data set are given in Fig. 5a, b. The corresponding distributions of $P_i$ values are given in Fig. 5c, d and do not show statistically significant heterogeneity. This is expected as homogenous GPs are normally observed in these GUV types, with little difference in the hydration across the bilayers. The distributions of GP values acquired from each data type is depicted in Fig. 5e. The average GP value of DOPC is shifted by roughly 0.4 units compared to DPPC:Cholesterol. Thus, in these model bilayer mixtures where spatial changes in hydration, and hence order, are not expected, the analyses does not detect domains.

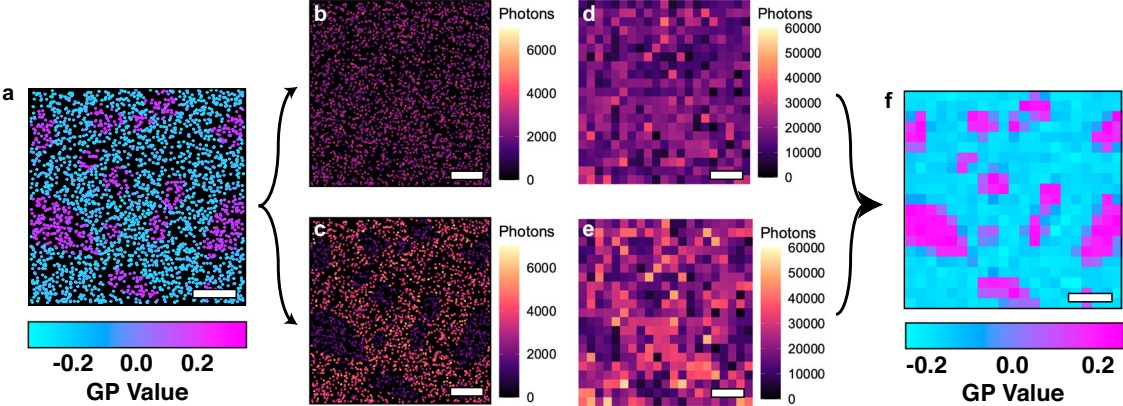

**Fig. 3 | Super-resolution rendering of simulated marked point pattern data using MOM. a** Simulated ratiometric marked point pattern colour coded for GP value. The data here has clear separation in the GP values, with higher GP values within domains. **b** and **c** represent the same point pattern from **a**) but now the points are assigned marks according to their simulated photon yields in the two channels. In **d** and **e** the marked point pattern is spatially binned (pixel size = 50 nm) and the photons of the detected points within those bins summed, to generate a sub-diffraction rendering of the photons detected within the acquisition. **f** These two images can then be used to calculate a super-resolution rendering of the GP within the region. Scale bars –200 nm.

## Measuring nanoscale membrane order in live cell membranes with PLASMA

Having demonstrated PLASMA on artificial membranes, we moved to probe the nanoscale membrane order of live cells. Ratiometric PAINT data was acquired on live rat mammary fibroblast cell (RAMA27) in the presence of excess di-4-ANEPPDHQ (80 nM) in DMEM (Fig. 6a) with a total imaging time of 8 min per field of view. Otherwise, the microscope setup and SMLM analysis pipeline were the same as for artificial membranes.

Data sets that showed heterogeneity via P-check were carried forward and analysed with JOSEPH and used to generate MOMs (Fig. 6b). For each domain identified, the mean GP of the domain was compared to the global average GP of its ROI to determine the absolute difference in GP values across domains (Fig. 6c, d). We observe mostly a central peak at 0 deviation from the global mean, with two relatively smaller lobes towards both negative and positive deviation (Fig. 6e). This suggests that there are 3 types of domains identified within the cell data: the central peak pertaining to domains with an equal mix of high and low ordered points, and domains comprising of higher proportions of higher ordered (positive deviation) or low ordered (negative deviation) points.

To demonstrate our method's sensitivity to the presence and changes in order we sought to perturb the membrane make-up of the plasma membrane. To do this we utilised a cholesterol binding molecule, methyl-β-cyclodextrin (MBCD), to remove cholesterol from the plasma membrane. Cells were treated with MBCD and compared to untreated cells using JOSEPH (Fig. 6c, d). P-Check identified heterogeneity in 49 of 57 ROIs from untreated cell data and 46 of 52 ROIs from MBCD treated cell data. We observed a decrease in the average GP value per ROI for MBCD treated cells when compared with the untreated cells (Fig. 6c), consistent with previous studies using this treatment. Upon inspection of the average deviation of domains from the average GP we observed that for both conditions in this case, domains now elicited a central peak and two smaller lobes relating to the lower and higher order domains identified (Fig. 6e). However, a relative enrichment in the number of low order domains is observed upon treatment with MBCD, suggesting that the distribution of domains as well as the global GP are becoming more disordered. Given that the domains identified are likely to be dynamic on living cells, we sought to determine how these dynamics may affect our ability to accurately segment domains (Supplementary Figs. 5 and 6). Using simulations, with parameters derived from our existing cell data (Supplementary Fig. 5a, b), we focused specifically on two scenarios: transient domain formation and lateral movement of stable domains

(Supplementary Fig. 5c–f). For transient domains, JOSEPH performed optimally when the domain was present for 25–30% of the acquisition time (~125–170 s). Additionally, we explored the drifting of stable domains within the data (Supplementary Fig. 6). While we could recapitulate domain shape well for small drifts, at high drifts (800 nm over the entire acquisition) the underlying ground truth is met less well, when the data is taken as a whole (Supplementary Fig. 6c–f). Although we are able to detect the higher order points within the region explored by the domain and detect it (Supplementary Fig. 6b), this does limit our ability to measure accurately its underlying size and shape due to the movement of the domain.

To determine if we can observe domain dynamics at these time scales, we implemented a sliding window approach on our untreated live cell data (Supplementary Fig. 7). Briefly, for each ROI, we take a window size of 2000 frames (i.e., 100 s, from frame 0-2000 as a starting timestamp), and shift the window by 10 frames, equivalent to 0.5 s. With this we can also calculate the domain areas for each segment, with 90% of all returned areas falling between $0.01 \mu m^2$ and $0.22 \mu m^2$ (Supplementary Movie 2). We can also analyse each window with JOSEPH (Supplementary Fig. 7a and Supplementary Movie 2) and then calculate the IoU between the initial window (frames 0-2000) and all subsequent windows (Supplementary Fig. 7b) to give a measure of the stability of the domain structure. To limit the potential of membrane structures, like protrusions, effecting the domain detection we tested all our data for spatial randomness (finding 43 out of 49 were spatially random), and used only ROIs that are not associated with the cell periphery. The IoU traces from all spatially random ROIs were averaged (Supplementary Fig. 7b). We observe the IoU drops to half its peak value within a 100-frame delay (equivalent to 5 s) in the sliding window. Furthermore, the IoU drops below 0.1 at a frame delay of 430 (equal to 21.5 s), suggesting a significant reduction in spatial overlap of domains within the width of one window. Overall, this suggests that dynamics of the domains detected in our data occur at timescales less than 2000 frames (100 s).

## Discussion

In this work, we have devised a series of analytic techniques for marked point pattern data. We highlighted their efficacy by implementing a ratiometric PAINT approach with the solvatochromic probe di-4-ANEPPDHQ to explore the distribution of membrane order, represented by GP values, at resolutions below the diffraction limit of light (Fig. 1). From such data we were able to detect and map the location of lipid domains in live cells.

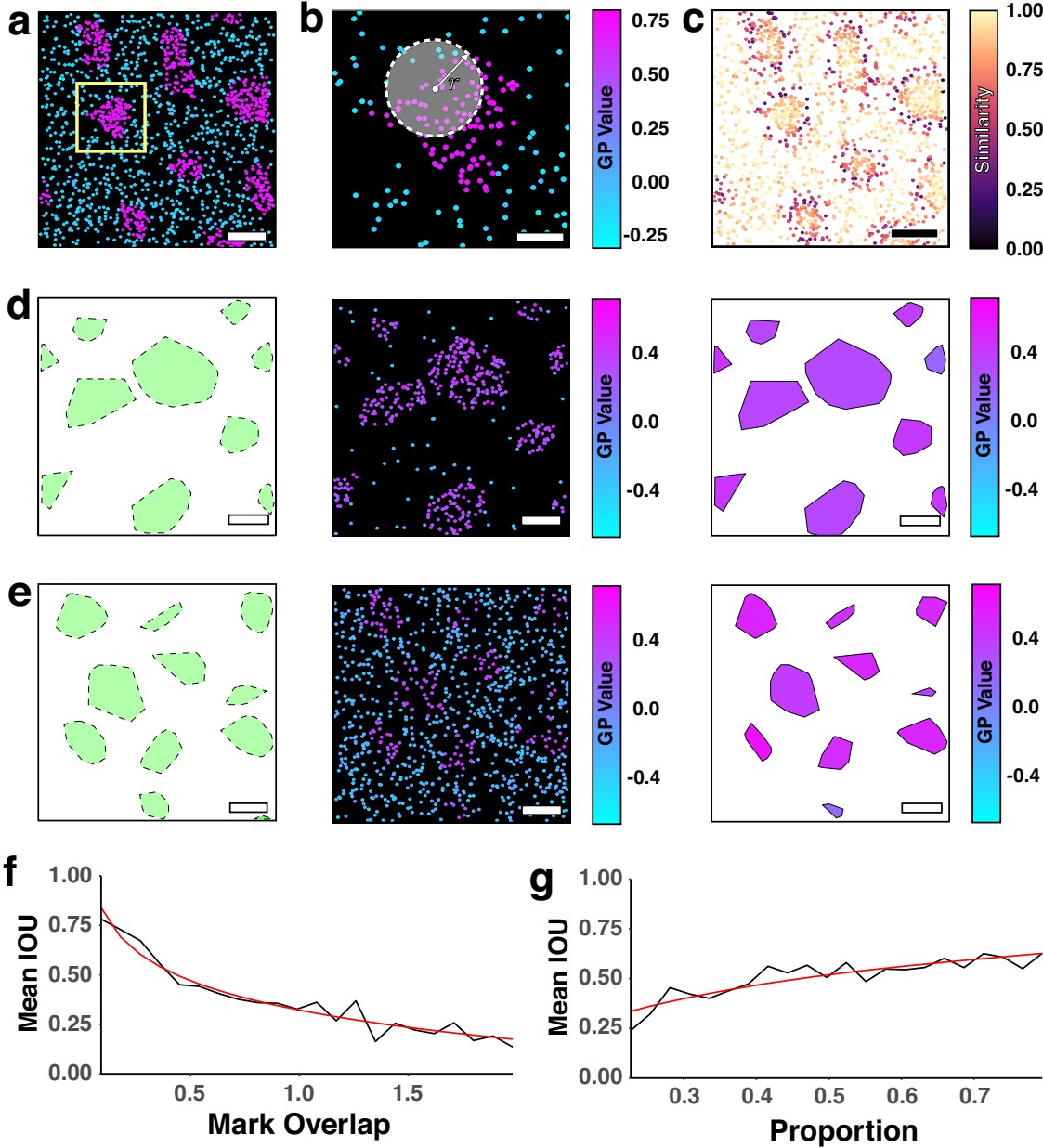

**Fig. 4 | Workflow and performance of JOSEPH on simulated marked point patterns. a** Simulated GP-marked point pattern containing high (magenta) and low order (cyan) domains. **b**) Zoom of the yellow highlighted region in **a**. **c** The GP value of each point is compared to its neighbours and assigned a local similarity score. Points on the boundaries of clusters typically display the lowest similarity scores. Clusters are constructed by iteratively attaching points to neighbours of high local similarity until a deviance threshold is met (here 0.2). JOSEPH performance on **d**) a spatially clustered distribution and **e**) a spatially random distribution, both with mark overlaps of ≈ 0.5. In (**d** and **e**) the convex hulls of the ground truth high order domains are shown (green polygons - right), and the underlying marked point patterns (centre, colour-coded according to GP value). Convex hulls determined by JOSEPH are shown (right), colour-coded with the average GP value of the points within the domain. **f** The graph of IoU versus GP mark distribution overlap (logarithmic trendline - red). **g** The graph of IoU vs proportion of points in domains (logarithmic trendline - red). JOSEPH was tested on 1000 simulated data sets and was repeated 3 times. Scale bars−panel (**b**) = 200 nm, all other panels = 500 nm.

The domains identified in live cells exhibited mean GP values that deviated significantly from the global average of their host ROI. This suggests the presence of ordered domains within disordered ROIs and disordered domains within ordered ROIs (Fig. 6). Ultimately, this could suggest a degree of regulated heterogeneity across the plasma membrane. As these scales are typically observed below the diffraction limit, this form of analysis offers a unique insight into membrane heterogeneity and the lipidome, in this case. Further, we were able to detect and quantify changes in membrane order and domain proportions upon manipulation of the plasma membrane with methyl-β-

cyclodextrin, where an increase in the disordered domain population was observed.

Only two input parameters are required for the analysis: a search radius (here, 50 nm) and the mark deviance threshold. While these may seem difficult to ascertain without evidence from the context, we have supplied additional methods for parameter estimation which are built into the software package by default (see Supplementary Fig. 2). Additionally, JOSEPH relies on the local mean as a measure of similarity between points – and this may not be the most accurate summary statistic for data sets of particularly low density, i.e., PAINT data where

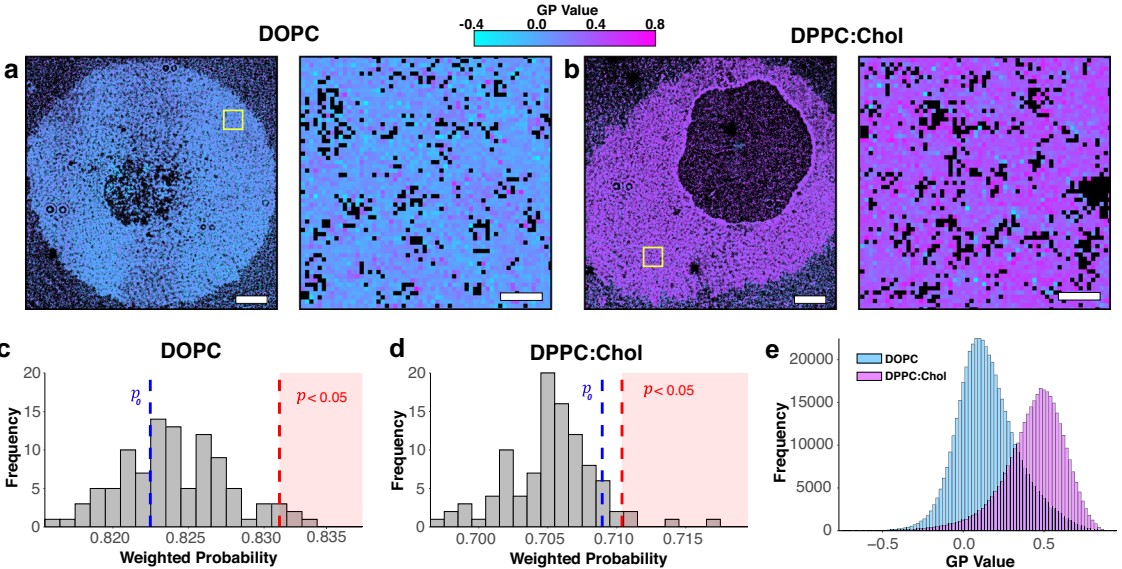

**Fig. 5 | Determination of membrane order and organisation in model membranes using di-4-ANEPPDHQ PAINT and PLASMA. a** Representative GP MOM of (**a**) DOPC and **b**) DPPC:Cholesterol membrane patches from a full-length acquisition (~8 mins). The right panel in each is a zoomed-in view of the yellow ROI in the left hand panel. Pixels in the MOMs are colour-coded according to GP, with black pixels denoting regions with no detected localisations. **c** and **d** are the P-Check weighted probabilities for DOPC and DPPC:Cholesterol membrane patches, respectively. P-Check does not find any statistically significant evidence of separation for either membrane type, as expected. **e** Histogram of the calculated

GP values from individual localisations for DOPC (cyan) and DPPC:Cholesterol (magenta), showing a distinct shift in the measured membrane order between the two model membrane mixtures. P-Check uses a one-sided permutation test with Bonferroni correction to adjust for multiple comparisons. The estimated *p*-values for Fig. 5c, d were 0.68 and 0.11, respectively. GUV data come from two independent experiments' data and 179 ROIS (DOPC: 93; DPPC: 86) were taken for analysis. P-check analysis was repeated 3 times on the GUV data. Scale bars - 5μm in left hand images, and 500 nm in zoomed regions.

we do not approach complete coverage of the membrane (with a density of 1 dye/584 nm², approximately, considered here as full coverage). Similarly, membrane structures (e.g., protrusions) could lead to artificial enrichment of detections in some areas. To limit this contribution, we tested all our data for spatial randomness and avoided the cell edges. However, if observation of membrane topography and order are desired, the PAINT approach demonstrated here could be combined with techniques such as MIET, to yield nanometre axial resolutions[30] and lifetime measurements of probe binding[31].

Fortunately, SMLM imaging, and particularly PAINT, can be exploited to provide extremely dense data sets, so it is easy to overcome this limitation[32,33]. However, it should be noted that whilst we can on average attain these densities, this comes at the cost of longer imaging times, in our case acquisitions were 8 min in length, which could miss short-lived dynamics within the data, or over-represent domain areas due to their potential mobility. Furthermore, this may also affect the resolution of domains, and resolution was not determined on live cell data due to these potential dynamics. To determine if we can observe domain dynamics at these time scales, we implemented a sliding window approach on our live cell data (Supplementary Fig. 7). On the timescales of the current acquisition, we observed that membrane order domain structure likely happens at time scales shorter than 100 s within our data. Given PAINT is a versatile approach for membrane imaging, we anticipate that chemical methods to alter the binding rates of solvatochromic probes could significantly improve acquisition lengths[15]. Such probes with narrower emissions (di-4-ANEPPDHQ has broad emission) would also allow multiplexing, and the ability to probe other molecules (e.g., proteins), or properties of interest (e.g., membrane charge or tension)[20], in relation to the membrane order, giving insights into the control of these by the lipidome.

In summary, we have demonstrated that marked point pattern data can be extracted from the solvatochromic probe di-4-ANEPPDHQ using a 2-channel ratiometric PAINT SMLM acquisition. When combined with statistical cluster and colocalization analysis, this can serve

as a basis for probing and mapping the nanoscale spatial organisation of ordered lipid domains in the cell membrane. Our data suggest that large scale segregation of domains may not be commonplace, but that ordered domains can be observed segregated from disordered areas at the nanoscale. Whilst we have demonstrated the potential of the methodology for membrane order mapping, this lays the groundwork for analysis of other marked point patterns acquired by SMLM.

## Methods

### Giant Unilamellar Vesicle (GUV) preparation
The preparation of GUVs was carried out by means of electroformation, which has been described previously[34]. All lipids were sourced from Avanti Polar Lipids. In brief, a lipid film was formed by depositing a chloroform solution of either DOPC (1,2-dioleoyl-sn-glycero-3-phosphocholine, cat.no: 850375 P) or a mixture of DPPC (1,2-dipalmitoyl-sn-glycero-3-phosphocholine, cat.no: 850355) with 30% (mol/mol) cholesterol (cat.no: 7000) onto indium tin oxide (ITO)-coated glass slides. In total the amount of deposited lipid was 100 μg. Subsequently, the lipid films were dried under a fume hood for 4 h. GUVs were formed in a 200 mM sucrose solution under the following conditions: 50 °C, 11 Hz, 1 V of alternative electric current for 2 h.

### Artificial membrane patch preparation
A 1 in 200 dilution of GUVs was made in 1xPBS and added to a well of a Ibidi μ-slide 8-well glass bottomed chamber. The GUVs settle in solution and in contact with the glass, under osmotic pressure of the 1xPBS, burst to form a patch of lipid bilayer.

### Rat mammary fibroblast cell culture and sample preparation
Rat mammary fibroblast cells (RAMA27), a gift from Prof. David G. Fernig (University of Liverpool), were cultured in plastic in phenol red free Dulbecco's modified Eagle's medium (Life Technologies, cat.no: 31053028) supplemented with 10% (v/v) foetal bovine serum (Life Technologies, cat.no: 16140071), 50 ng/mL insulin (Sigma Aldrich,

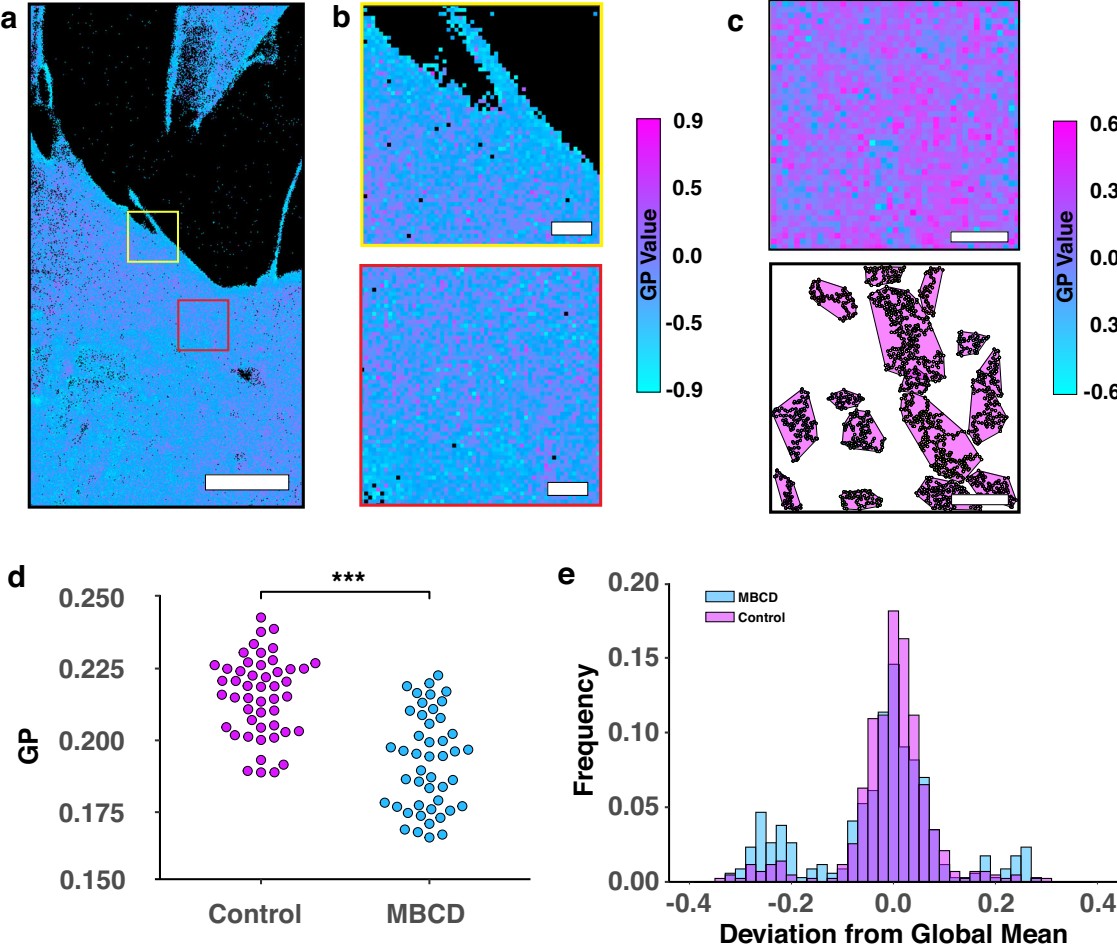

**Fig. 6 | Quantifying changes in membrane domains and order in live cells using di-4-ANEPPDHQ PAINT with PLASMA. a** Representative MOM of live RAMA27 cell membrane (left) from a full-length acquisition (~8 mins), with zoomed-in views of the yellow and red ROIs (**b**). Pixels in the MOMs are colour coded according to GP value, with black pixels denoting regions with no detected localisations. **c** Exemplar region of di-4-ANEPPDHQ PAINT live cell data, reconstructed using MOM (top). Domains whose average GP values were significantly different to the global mean (*i.e.* average GP of all marked points in the ROI) were segmented by JOSEPH (bottom). Points belonging to the ordered domains were rendered with different colours, and the convex hulls were colour-coded according to the average GP value domain. **d** Average GP value for all points within selected ROIs for untreated (Control, magenta) and methyl-β-cyclodextrin (MBCD, cyan). Statistical significance was determined via two sample t-test over 57 control ROIs and 52 MBCD ROIs. Significance ranking: n.s - not significant, * $p < 0.05$, ** $p < 0.01$, *** $p < 0.001$. **e** Histogram of the frequency of domains identified by JOSEPH fell into intervals above (*ordered*) or below (*disordered*) the global average GP value (here, -0.22 for untreated control and -0.17 for MBCD) for both the untreated cells (Control, magenta) and those treated with methyl-β-cyclodextrin (MBCD, cyan). 109 ROIs from two independent experiments (total of 6 cells) were used for analysis (57 untreated ROIs, and 52 MBCD ROIs). The test for Fig. 6d was a two-sided t-test with one comparison made, so no adjustments were used. p-value for Fig. 6d was $10^{-6}$. Scale bars: **a**) 5 μm in left hand large FOV, and 500 nm in **b**).

cat.no: I0516), 50 ng/mL hydrocortisone (Sigma Aldrich, cat.no: H0135) at 37 °C in 10% (v/v) $CO_2$[35]. When required for imaging they were trypsinized and seeded into Ibidi μ-slide 8-well glass bottomed chambers, pre-coated with 1 μg/mL fibronectin (Sigma Aldrich; F4759), at 5,000 cells per well and left overnight in culture medium to adhere and spread, before PAINT imaging.

### RAMA27 treatment with methyl-β-cyclodextrin
Seeded cells were exposed to 15 mM methyl-β-cyclodextrin (Sigma Aldrich, cat.no: C4555) in culture medium without foetal bovine serum for 30 min. The cells were then washed 3 times with 1xPBS to remove the methyl-β-cyclodextrin and prepared for imaging.

### Ratiometric PAINT imaging with di-4-ANEPPDHQ
Samples were incubated with 20 nM di-4-ANEPPDHQ (Life Technologies, cat.no: D36802) in 1xPBS buffer (artificial membranes) or 80 nM di-4-ANEPPDHQ in full culture medium (live cells, see ***Rat mammary fibroblast cell culture and sample preparation***) to allow

binding of single di-4-ANEPPDHQ molecules for PAINT image acquisition. Concentrations were tuned using the samples to achieve high localisation coverage of the samples, i.e., at least one localisation per 584 nm² in the final data. The authors would advise that this optimisation is done in house for every sample. Acquisition was performed on a custom-built fluorescence microscope (RAMM system, ASI) using a 100× 1.49 NA oil immersion objective (Apo TIRF; Nikon) and a beam shaper for homogeneous illumination (piShaper, AdlOptica). Hardware was controlled using μManager 2.0 extended with custom microcontroller boards. 10,000 frames with an integration time of 50 ms were acquired at 50 W/cm² using 488 nm laser excitation under HiLo illumination and z-focus lock (CRISP, ASI). Fluorescence emission from binding of di-4-ANEPPDHQ to membranes was split onto two synchronised sCMOS cameras with 130 × 130 μm field of view (Prime 95B, Photometrics) via a LP640 dichroic mirror (FF640-FDiO2-t3, Semrock). Fluorescence < 640 nm is further filtered using a custom bandpass filter (552/96). Both image stacks are retained for further analysis.

**Table 1 | Input parameters for simulated data sets and the range of values they were randomly generated from**

| Parameter | Function | Min | Max |
|---|---|---|---|
| $\mu_1$ | Mean mark value in simulated domains. | 0.01 | 1 |
| $\mu_2$ | Mean mark value outside simulated domains. | −1 | −0.01 |
| $\sigma_1$ | Standard deviation of marks in simulated domains. | 0.01 | 0.5 |
| $\sigma_2$ | Standard deviation of marks outside simulated domains. | 0.01 | 0.5 |
| $p$ | Proportion of points assigned to domains. | 0.1 | 0.9 |
| $n$ | Number of points in marked point pattern. | 200 | 2000 |
| $p_t$ | Percentage of total time in which domain is visible. | 1 | 100 |
| $d$ | Simulated lateral drift (in nm/frame). | 0 | 8 |

## SMLM data processing and ratiometric analysis

Image stacks were initially combined vertically, positioning the channel with the long-pass emission band (greater than 640 nm) at the top, and the short-pass emission band (505-600 nm) at the bottom. All processing up to GP calculation was carried out in SMAP/MATLAB[36]. We adhered to the step-by-step guide for dual-colour data by Ries et al. with minor changes. Initially, localisations were determined using the 'fit_fastsimple' workflow. During this process, camera-specific parameters were employed, including gain (0.52 e-/ADU for Sensitivity or 1.07 e-/ADU for Balanced), offset (100), pixel size (111.4 nm), exposure (50 ms), and time difference (50 ms). These were combined with a Difference of Gaussian (DoG) peak finder (1.2), a dynamic factor of 1.3 (Balanced) or 1.1 (Sensitivity), and the 'PSF free' fitting approach with enabled asymmetry detection.

After localisation, a filtering of the localisations was implemented, based on localisation precision (0–50 nm), PSF width (100–300 nm), frame range (400-Infinity), and asymmetry (0-0.2). Next, the projective transformation between the two channels was calculated, designating the bottom channel as the target and the top channel as the reference. This registration process utilised a pyramidal approach to identify matching landmarks (Pixel sizes: 1000, 500, 250, 100, 50, 25; Max Shifts: 2500, 1250, 625, 250, 125, 63).

Following this, dual channel intensities were assigned by reanalysing the image stacks and determining the number of emitters in both the target and reference channels. The background was estimated using the 40th quantile from 20 frames and used for background subtraction. For peak fitting, the 'roi2int_fitG' workflow was selected with an ROI size of 7.

Localisation events of Di-4-ANEPPDHQ were distinguished from fiducial markers through ratiometric intensity filtering. Events with more than 5000 photons in either channel were classified as fiducials, while those with 50–5000 photons in each channel were identified as PAINT localisations. Subsequently, localisations from the bottom channel were mapped onto the first, utilising the transformation matrix generated during registration. After transformation, the localisations underwent another filtering step (localisation precision: 0–50 nm; PSF: 100–250 nm; frame range: 400-Infinity; asymmetry: 0-0.2) before a drift correction was performed at 10 nm pixel size and 10 timepoints using a smoothing cubic spline interpolation.

Localisations were grouped (fixed 50 nm, dT: 1), and any grouped localisation events with fewer than 100 photons in either channel were excluded from further analysis. Subsequently, the GP value was calculated from the integrated photon numbers derived by fitting in each channel according to Eq. (1):

$$GP = \frac{Photons_{505-600nm} - Photons_{>640nm}}{Photons_{505-600nm} + Photons_{>640nm}} \quad (1)$$

## Marked point pattern data simulations

Over 2000 point clouds were simulated using agent-based modelling techniques and overlaid with mark values acquired from two arbitrary Normal distributions[37]. Input parameters (described in Table 1) were randomly selected via Latin Hypercube Sampling[38]. Furthermore, a subset of 1000 data sets was generated as completely spatially random (CSR). Each simulation was 3 μm × 3 μm in size and fitted with convex domains of varying geometries. Marks were assigned so that the average value of points inside domains was positive, and the average outside was negative.

## Domain lifetime simulations

1000 simulations were each run over 10000 simulated frames within a 3 μm × 3 μm ROI. At each frame, a random number of CSR points were generated in the ROI. This number was pulled from a discrete probability distribution, estimated from the number of localisations acquired in each frame within experimental data (Supplementary Fig. 7a, b). A central circular domain of radius 100 nm was included for a percentage $p_t$ of the simulated acquisition time, initialised at frame 1. 10 random trials were generated for each integer percentage between 1% and 100% inclusive. Any points generated within the circular domain, during the percentage of time for which it was included, were assigned a GP value from a Normal distribution with mean $\mu = 0.2$ and standard deviation $\sigma = 0.1$. All other points were assigned a GP value from a Normal distribution with mean $\mu = -0.2$ and standard deviation $\sigma = 0.1$. Domains were recovered with JOSEPH, and IoU scores (see below) were calculated by comparing recovered hulls with the convex hull of the underlying ground truth domain.

## Domain drift simulations

900 simulations were each run over 10000 simulated frames within a 3 μm × 3 μm ROI. Spatially random points were overlaid in each frame with point generation probabilities taken as above. A drifting circular domain of radius 100 nm was initialised with centre at coordinates (100, 1500). The x-coordinate of the centre of this domain was offset at each frame by a given number of nanometres, defined by a lateral drift parameter $d$. 100 random trials were generated for each integer value of $d$, between 0 nm per frame and 8 nm per frame inclusive. Any points generated within the circular domain were assigned a GP value from a Normal distribution with mean $\mu = 0.2$ and standard deviation $\sigma = 0.1$. All other points were assigned a GP value from a Normal distribution with mean $\mu = -0.2$ and standard deviation $\sigma = 0.1$. Domains were recovered with JOSEPH, and IoU scores (see below) were calculated by comparing recovered hulls with the convex hull of the underlying ground truth domain at its initial position (100, 1500). Circularity was calculated as $4\pi A/P^2$, where $A$ is the area and $P$ is the perimeter of the recovered hull.

## P-check

The method itself operates on the premise that data sets exhibiting non-random colocalization of identically labelled points will express, on average, neighbourhoods which are predominantly homogeneous. We initialise the algorithm with a search radius, $r = 50$ nm (determined by Ripley's K analysis, Supplementary Fig. 2), from which we can determine the neighbourhood of each point. We then iterate over each point, recording the total number of neighbours and the proportion of which share a mark value with the original point (Fig. 2a–c). These values can be summed over all points to determine the weighted probability that a randomly selected point and any randomly selected neighbour share the same label. This is expressed mathematically as,

$$P_0 = \frac{\sum_{i=1}^{n}\sum_{j=1}^{n}\mathscr{H}\left(|x_i - x_j| - r\right)(1 - \mathscr{H}(|l_i - l_j|))}{\sum_{i=1}^{n}\sum_{j=1}^{n}\mathscr{H}\left(x_i - x_j - r\right)} \quad (2)$$

where $\mathbf{x}_i$ is the spatial coordinate of point $i$, $l_i$ is the associated label of point $i$, $n$ is the total number of points and $\mathscr{H}$ is the Heaviside function, which equates to 1 when the input is positive[39].

The $P_0$ value will be maximised if points belonging to the same category form distinct, spatial clusters–in particular, if there is no mixing between points of different categories and clusters of each category are well-separated. Once $P_0$ has been calculated, a permutation test is performed. This is done by shuffling the categorical labels of all points randomly and recalculating the $P_i$ value over a given number of iterations. If the original $P_0$ falls within the top $\alpha\%$ of all $P_i$ values derived from the permutation test, then this suggests that there is statistically significant evidence at the $\alpha\%$ level of non-random colocalization of identically marked points[40].

To test the applicability of P-Check, we simulated 1000 distributions from which each mark was derived from one of two possible Normal distributions. Each point pattern was discretised using Gaussian mixture modelling to recover the original point of intersection of the two Normal distributions[23]. This value was used as a threshold to separate points into one of two categories (representing liquid and gel phase respectively)[41]. Statistical analysis reveals that the $P_0$ value decreases as the overlap, defined by,

$$o = \frac{\sqrt{\sigma_1 + \sigma_2}}{|\mu_1 - \mu_2|}, \tag{3}$$

increases (Fig. 2e). Examples of P-Check's performance on a clustered and CSR data set (with low and high overlap respectively) are given in Fig. 2f, g.

**JOSEPH**

A "neighbourhood" comprised of all other points within the chosen search radius (here, 50 nm) is constructed. Each point is then designated a similarity value defined as the difference between the point's own mark and the average mark of its neighbourhood. This value is normalised so that points which are most like their neighbourhood are given a similarity value ranging from 0–1, *i.e.*, lowest similarity and highest similarity, respectively. Points are then ordered from highest to lowest similarity value. By iterating over the filtration in order and connecting points to other points of lower filtration value (higher similarity) spatial clusters containing points which share comparable marks to their neighbours are generated. A threshold is used to prevent merging of all points within the data, denoted $\tau$, which is the maximum acceptable difference between the marks of the root (starting point) and any other point in the cluster. After each iteration points were connected to all neighbouring points whose similarity values are higher than its own, and whose roots' marks are at most $\tau$ higher in value. In cases where two separate clusters bridge, the cluster whose root had the lowest similarity value is combined with the cluster possessing the higher root. The convex hull of the proposed clusters was used as an estimate of the domain boundary. The method was also extended to allow dynamic observation of domain structure in synthetic and experimental data. This was achieved by comparing an initial window (2000 frames here) to all consecutive windows (of equal size, i.e. 2000 frames), separated by 10 frames (0.5 s). The same JOSEPH analysis was done on the subsets (Supplementary Fig. 7). In all cases, IoU scores were calculated between pairs of hulls by taking both the intersection and union of the areas of all hulls, then dividing the area of the intersection by the area of the union.

**Membrane order mapping (MOM)**

The x, y, and integrated photon intensities for each channel were taken from the processed ratiometric PAINT data. These data were used to generate a 2D histogram (bin size = 50 nm) of the sum of the photons for all the localisations that resided within each bin. This process was done separately for the 505–600 nm channel and > 640 nm channel

photon numbers, thus generating two 2D histograms of the summed photons for each channel respectively. These two summed photon images were then used to calculate the GP value per 50 nm bin, according to previously described GP image methods[7]. Briefly for each pixel the operation was as follows;

$$\text{GP bin value} = \frac{\sum \text{Photons}_{505-600\text{nm}} - \sum \text{Photons}_{>640\text{nm}}}{\sum \text{Photons}_{505-600\text{nm}} + \sum \text{Photons}_{>640\text{nm}}} \tag{4}$$

This yielded a sub-diffraction representation of the local GP values within the data.

**Reporting summary**

Further information on research design is available in the Nature Portfolio Reporting Summary linked to this article.

## Data availability

Simulated data sets and experimental data used in this work are available without restriction at https://github.com/lucapanconi/PLASMA. Source data are provided with this paper.

## Code availability

The PLASMA software[42] package v1.0.0 was written in the R programming language v4.2.2, and employed in the integrated development environment RStudio, version 2022.07.1 + 558. Software is freely available under GNU General Public License v3.0 (https://github.com/lucapanconi/PLASMA).

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

## Acknowledgements

L.P., D.M.O. and D.J.N. acknowledge funding from Oxford Nanoimaging (ONI) and the Engineering and Physical Sciences Research Council through the University of Birmingham CDT in Topological Design, grant code EP/S02297X/1. J.E., S.A.T. and D-P.H. received funding from the Academy of Medical Sciences (Grant APR2\1013). All authors acknowledge funding by the Centre of Membrane Proteins and Receptors (COMPARE, Universities of Birmingham and Nottingham).

## Author contributions

L.P. wrote simulation and PLASMA analysis code, produced simulations, and performed analyses. J.E. built the optical set-up, acquired di-4-ANEPPDHQ PAINT data, and performed ratiometric analyses. S.A.T. acquired di-4-ANEPPDHQ PAINT data and performed ratiometric analyses. M.M. prepared GUVs. D-P.H. contributed ideas and concepts. D.J.N. developed the di-4-ANEPPDHQ approach, optimised the generation of membrane patches, wrote the MOM code, and performed ratiometric analyses. D.M.O. and D.J.N. conceived the work. All authors contributed to the draughting and writing of the manuscript.

## Competing interests

The authors declare no competing interests.
