## [Transparent Peer Review file · Nature Communications]

Mapping membrane biophysical nano-environments

Corresponding Author: Dr Daniel Nieves

Version 0:

Reviewer comments:

Reviewer #1

(Remarks to the Author)

Uploaded as a pdf to preserve formatting

Reviewer #2

(Remarks to the Author)

The manuscript "Mapping membrane biophysical nano-environments" describes a method/software package to analyze membrane biophysical environment through changes of fluorescence emission of a solvatochromic probe (4-ANEPPDFQ) with so called "spectrally-resolved Single-Molecule Localisation Microscopy (SMLM)". By computing generalized polarization (GP) from the SMLM data, the author developed an algorithm that allows the detection of lipid domains (rather groups of similar GP spots) and validated the algorithm with both artificial membrane and live cell membrane. The cell membrane is of vital importance organelle to the cellular physiology and functions. Yet, there are limited tools to study the the dynamics and functional compositions and it is a hotly debated field. Ultimately, visualization of so-called lipid rafts would yield significant understandings of such basic cellular functions. This tool not only extend the spatial information of SMLM data but also added extra dimension of information of the labelling environment and it would be highly useful in the biomedical fields.

The manuscript is well written in an easily understandable format with nicely done illustrations. However, numerous sections of the text refer to wrong figure numbers/panels and some were no existent (page 7) which is maybe due to some carelessness in manuscript revisions. Nevertheless, the methodology is sound, and I particularly appreciate the automated estimations of the 2 key parameters (radius and the deviance) which is often problematic in the success of image analysis field where key parameters require often subjective estimations.

For the manuscript, I have the following suggestions:

- 1) As a biologist, I cannot help but wonder if any experimental perturbation is applied to the cell and any changes were detected in the distributions of domain sizes, numbers. This is maybe beyond the scope of the paper. But expanding the biology of the method would excite many biologists in the field. Too often, new methodology was developed. But it remains in the expert labs and failed adopt widely due to the lack of example applications.
- 2) The method sounds solid with examples of successes in detecting clusters of domains with similar GPs. I am wondering if some discussions on the limitations of the techniques could be expanded. For example, how robust is the estimations of radius and deviance? This is particularly useful for the biologists where many lack necessary mathematics or statistics background to know the limitations of the methods. The model is built onto the assumptions that either clusters or non-clustered spots follow a normal distribution (which is likely true in most of the cases). But what if it does not (even in figure 5, panel e shows a not so normal distribution pattern)? Additionally, the method of paint-like SMLM yield relatively dense point SMLM data pattern. What is the minimum level of numbers of points needed for the algorithm to succeed? This is somewhat addressed in the discussion ("this may not be the most accurate summary statistic for data sets of particularly low density. ..."), but a bit more than a "low density" would be appreciated because "low" is not a scientific term.

Some minor (but really annoying!) issues:

Page 3: 1st paragraph "(here, membrane lipid order) as a point of interest (Figure 1a-c).." It should be Figure 2?

Page 4: "Figure 1..." should be Figure 2?

Page 7 "domains will not be identified(.) Further, the proportion..."

Page 7 I am not sure where these figures refers to? "Figure 3g), i.e., if the mark values within spatially clustered points are similar to those outside of the clusters then domains will not be identified Further, the proportion of points assigned to domains (Figure 3h" But there is no panel 3h!

Page 7 "We first acquired data using established GUV preparations..." please define GUV.

Page 8 Whether it is figure 5 or 4? "Examples of two MOMs from each data set are given in Figure 5a and 5b. The corresponding distributions of $P!$ values are given in Figure 4c and 4d and do not show statistically significant heterogeneity. This is expected as homogenous GPs are normally observed in these GUV types, with little difference in the hydration across the bilayers. The distributions of GP values acquired from each data type is depicted in Figure 4e. The average GP value of DOPC is shifted by roughly 0.4 units compared to DPPC:Cholesterol. Thus, in these model bilayer mixtures where spatial changes in hydration, and hence order, are not expected, the analyses do not detect domains."

Page 8: "the presence of excess di-4-ANEPPDHQ (80 nM) in DMEM (Figure 5a)." Figure 5A or 6A? Also, (80nM)...This is 4x higher concentration than the artificial membrane experiment and I am just wonder why?

Page 9, Figure 6? In this paragraph? "MOMs (Figure 5a-c). For each domain identified, the mean GP of the domain was compared to the global average GP of its ROI to determine the absolute difference in GP values across domains (Figure 5c). Results show two clear prominent peaks comprising clusters with GP values significantly above and below the global average, respectively. This suggests the existence of low order domains occupying regions of high order and the existence of high order domains occupying regions of low order. Additionally, the area of the convex hull of each cluster was determined (Figure 5d),"

Supplementary information

1) I could not install the PLAMA with the follow Error:

"Failed to install 'PLASMA' from GitHub:

Failed to install 'RSMLM' from Git:

Could not find tools necessary to compile a package

Call `pkgbuild::check_build_tools(debug = TRUE)` to diagnose the problem."

2) The supplied movie failed to display on my office desktop with windows media player. Although I was able to check it on a Mac OS computer, please check the compatibility of the movie. Here is some info of my office computer:

Edition Windows 10 Enterprise

Version 21H2

Installed on 10/21/2022

OS build 19044.3208

Experience Windows Feature Experience Pack 1000.19041.1000.0

Reviewer #3

(Remarks to the Author)

In their manuscript "Mapping membrane biophysical nano-environments", Panconi et al. use single molecule localisation microscopy to investigate the presence of clusters with different lipid compositions within the plasma membrane. For this purpose, they employ the dye di-4-ANEPPDHQ, which exhibits different spectral properties depending on the surrounding lipid composition. According to the authors, the application of the dye from the outside leads to the integration of dye molecules into the membrane, creating a fluorescent point source for the use with SMLM algorithms to extract super resolution coordinates. The use of intensity data from two different color channels is used as an indicator for the type of lipid environment of the dye molecule (liquid disordered or ordered). This data is then used in the algorithms "P-check" and "JOSEPH" to detect the presence of clusters and their shape. The use of SMLM in combination with these algorithms allows to identify domains with higher order of lipid than would be possible with regular, diffraction-limited microscopy. The principles of the algorithms are first introduced and explored using simulations and later applied to vesicle preparations and living cells.

I very much like the concept of using di-4-ANEPPDHQ with SMLM. The ideas behind the authors' algorithms are clear and the implementation and documentation on their GitHub site is reasonably easy to follow. This technique harbours the potential to become an important tool for studying the structure of lipid membranes and the principle could be translated to other sensors. As of now, however, I feel that several of the underlying assumptions need to be explored further before the authors' conclusions are sufficiently supported by the data. This is particularly necessary as the application of the authors' algorithms to cellular samples as shown in Fig. 6 is not fully convincing to me yet.

1) Are the chosen concentrations of di-4-ANEPPDHQ appropriate? My impression from the accompanying video is that the density of signals is rather high and it may be common to record overlapping emissions. It might be worthwhile to test several concentrations. Lower concentrations can also be helpful to better narrow down the number of photons from a single emission event (50-5000 photons sounds like a rather large range). Do the GP values change with concentration? There is a report (<https://pubs.acs.org/doi/10.1021/acs.jpcc.0c09496>) stating that this may be the case in cellular but not synthetic membranes.

2) The contribution of background noise must be determined for each color channel. This should be done in the absence of sample and/or dye. In particular for the cellular samples shown in Fig. 6, this must also be done with cells but without dye as several endogenous molecules may create endogenous signals.

- 3) How exactly do you deal with chromatic aberration (CA) and sample drift? In particular, accurate correction of CA is crucial to calculating the GP value for an individual dye emission.
- 4) Connected to the above: What is the true resolution of your setup? While SMAP may be able to determine to determine each individual localization with 0-50 nm precision, after CA, drift correction, and grouping, the uncertainty propagates and may be much higher than assumed. Is the 50 nm search radius really appropriate for the resolution of the data?
- 5) Is the GP value only dependent on the surrounding lipids? What about the presence of (trans-)membrane proteins?
- 6) May the GP value be affected by the orientation of the membrane/dye relative to the incident light? You assume a perfectly flat surface in your simulations but real samples may be unevenly attached. For example in Fig. 5b, it seems as if "low GP spots" are associated with black spots (without signal, perhaps because they are not in the plane of excitation?).
- 7) I presume that the localisations in your simulations are stationary but this may not be the case in cellular samples. Can you estimate if lateral drift of lipids (and perhaps even full domains) occurs during your recordings? Perhaps it is possible to split the recordings into multiple sub-stacks and check if the features are stationary over time or not?
- 8) Is MOM really required and can the same result not simply be achieved by rasterizing the original GP point pattern (i.e. binning Fig. 3a)? Also, you state that MOM is explained in "Methods and Materials", which it is currently not.
- 9) It seems to me that the underlying assumption of the manuscript is that there is a clear cutoff between high and low order domains. Is this assumption valid? The domains in Fig. 6 appear much less clearly defined. Do you find differences within the domains regarding GP value "density"? If present, maybe these could be analysed as well?
- 10) It would be a good control for the applicability to real samples, if you could demonstrate (dynamic) changes to the lipid composition of your model system. Perhaps depletion of cholesterol in your cellular samples using methyl- β -cyclodextrin could be performed?

Dr. Gabriel Stölting
Center of Functional Genomics
Berlin Institute of Health at Charité - Universitätsmedizin Berlin

Version 1:

Reviewer comments:

Reviewer #1

(Remarks to the Author)

I appreciate the authors' effort to make the paper clearer and eliminate all the original errors in the text and figures. Now that I can understand the paper better, I still have some concerns and suggestions.

1. The authors did not address my major concern adequately. They now correct the typo on the single molecule localization, but it is still small, at 23nm, on the scale of diffusion within the time per frame of 50ms. These small organic molecules would diffuse out of the 23nm region in less than 1ms, if my estimate of a $\sim 1\mu\text{m}^2/\text{s}$ diffusion constant is close to correct. How can such a fleeting molecule be localized with such precision within 50ms? Do the authors have some reason to believe that my estimate of the diffusion is way too high? It would be interesting to do a simple FRAP experiment on di-4-ANEPPDHQ labeled cells or GUVs to see what D is. Perhaps the localizations represent a sub-population of dye molecules that are forming less mobile aggregates in the membrane; such aggregates could have altered spectral properties and compromise the interpretation of the data in terms of lipid order. Incidentally, the original work by Sharonov & Hochstrasser, 2006, used immobile lipid vesicles as $<100\text{nm}$ targets for Nile red; so the diffusion of the dye within these spatially restricted membranes was not a concern.

2. The time for an experiment is about 8 minutes, if I understand correctly (this should be made clearer to readers right in the Figure legends for Figs. 5 and 6). This could be a real limitation for any practical application of the method as small lipid domains are likely to have much faster turnovers.

3. It would be valuable to add something to the discussion about what specific cell biological questions might be addressed with this methodology.

Reviewer #2

(Remarks to the Author)

The authors addressed all my concerns and I think adding the m-beta-Cyclodextrin treatment experiments strengthened the methodology. I have no objection for the publications as it is.

Minor issues:

Figure 1 (should be 2!) (page 4) was still not labelled correctly in the merged document which is the one I read.

The merged file contains 2 copies of the main manuscript. The authors (did all of them?!) should read it carefully.

Reviewer #3

(Remarks to the Author)

I would like to thank the authors for considering and responding to my comments. Overall, the manuscript has greatly improved and almost all of my issues have been satisfactorily addressed. However, in my view, there are two remaining problems with the answers to my previous comments #7 and #6 that require further consideration before the author's claims are sufficiently supported. I pasted my original comments and the author's replies below and put my novel comments underneath.

My original comment #7 and the author's reply was as follows:

"7) I presume that the localisations in your simulations are stationary but this may not be the case in cellular samples. Can you estimate if lateral drift of lipids (and perhaps even full domains) occurs during your recordings? Perhaps it is possible to split the recordings into multiple sub-stacks and check if the features are stationary over time or not?

a. We agree with the reviewer that its certain that the lipids will be moving in all settings (artificial and live cell membranes), thus domains themselves may move. As suggested, we have now included subsampled example of the MOM allowing the potential movement domains to be observed (Supplementary Figure 5). Given this is it still difficult to make these estimates, as the diffusion of lipids is still at a much quicker scale than the sub-setted data (approximately 2 min windows bound by the need for a minimal density of localizations as described previously to perform the analysis)."

I would like to thank the authors for analyzing the sub-setted data as suggested. While I am certain that individual lipids will be moving rapidly, I don't necessarily assume the same for the lipid domains which are the object of investigation in this manuscript. There are likely several theories on the organization of these domains but I find it conceivable that they are (at least partially) tethered to membrane proteins and thus much less mobile than individual lipids.

Unfortunately, I find the results worrisome. It is hard to say from the representative images in Sup. Fig. 5 alone but it appears that the domains are not stable over the typical duration of the recordings used in the manuscript (8,000-20,000 frames accordings to Materials & Methods). This is also what I take from the author's answer to my comment. Should the domains even move significantly within the stated minimum window of 2 minutes, it would be even more problematic.

If these limitations were true and the temporal resolution of the recording system is much below the movement speed of the observed lipid domains, the resulting maps are mere averages with a lower resolution than the stated "nanometer(s)". Using short and standardized recording protocols will clearly minimize this error yet the actual resolution will be unknown and may be different for each sample.

I strongly feel that this issue needs to be investigated in more detail beyond the representative example in Sup. Fig. 5. I'm not sure if it would be worth the effort but simulations that include movement of the lipid domains may help to determine limits for the algorithms. Such simulations may also help to further optimize the minimum number of frames required. Also, it would be helpful to determine the speed of movement of the lipid domains. If feasible, one could use a moving analysis window (of 2 minutes or lower, if possible) and define the clusters using JOSEPH. This may allow to visualize and quantify the behavior of these domains over time.

My original comment #6 and the author's reply was as follows:

"6) May the GP value be affected by the orientation of the membrane/dye relative to the incident light? You assume a perfectly flat surface in your simulations but real samples may be unevenly attached. For example, in Fig. 5b, it seems as if "low GP spots" are associated with black spots (without signal, perhaps because they are not in the plane of excitation?).

a. This is a good point regarding dye orientation. The reviewer rightly points out that we have dark spots in the data, which are due to low coverage, i.e., no localisations within the pixel bin, and could arise from dyes not being optimally in the plane of excitation for live cells.

However, even in this case it's unlikely the GP value will be affected for the dye, i.e., the ratio of the short and long channel, but more the overall intensity of emission. This means that in area where we may have sub-optimal excitation and at the extremes of the GP scale, we may filter these events due to our stringent filtering to preserve the quality of the localisations used for analysis, i.e., rejecting localisations with very low photon numbers, e.g., like those seen on the slip or cells only."

I generally agree with the authors that the filtering and ratioing does remove a lot of low-quality localizations. The improved description of the settings used for filtering is highly appreciated. However, I do have remaining doubts as to the issue of a non-flat membrane surface. As the authors are aware, the inherent axial chromatic aberration of the objective results in a slightly different focal plane for each color. Shifting the position of a fluorophore in the z-axis may thus differentially affect the

signal from the two color channels. I presume that this affects all experimental recordings to some degree as the membranes (synthetic or cellular) will rarely be perfectly flat and contributes to the overall noise of the system. However, I remain worried that this results in a systematic error in some areas such as near the border of cells or around larger deviations from a flat membrane (due to artifacts but perhaps also vesicular structures). If this were the case, this might be detected by having systematically different PSF sizes near these structures.

Dr. Gabriel Stölting
Center of Functional Genomics
Berlin Institute of Health at Charité - Universitätsmedizin Berlin

Version 2:

Reviewer comments:

Reviewer #1

(Remarks to the Author)

The authors have provided a clarification in their response to my main concern. I thought they were claiming 23nm resolution, but they are actually reporting 1 localization per 23x23 nm region. I now understand that they are referring to the density of localizations rather than the resolution. While having a high density of localizations is a prerequisites for attaining good domain resolution, it is by far not the only factor. The authors claim nano-scale resolution, but only demonstrate this in their simulations. They need to demonstrate this with experimental data, containing real measurement noise and probe diffusion. They keep referring to their method as detecting "nano-environments", but have not established the resolution of their measurements.

I am satisfied that they now more clearly state the slow acquisition time required for the measurements.

I am satisfied that they have added some potential biophysical questions that might be addressed with the method.

Reviewer #3

(Remarks to the Author)

The authors have significantly improved the manuscript and the potential but also the limitations of the method are now much clearer. I would be very happy to see this paper being published in Nature communications and I'm eagerly looking forward to see how the authors and other groups will employ the developed algorithms in the future.

Dr. Gabriel Stölting
Center of Functional Genomics
Berlin Institute of Health at Charité - Universitätsmedizin Berlin

To the Reviewers,

Firstly, we would like to thank all the reviewers for their valuable feedback, and we sincerely apologise for the error in Figure numbering, which we acknowledge would have made the review difficult. Nonetheless we thank the reviewers again for persevering and giving their comments.

Below we respond to each of you in turn (by number), addressing your specific points (in red):

Reviewer 1

This paper is potentially of value as it introduces a method for determining and analyzing nanoscale lipid domains in cell membranes. The methods build upon established approaches that have been demonstrated at lower resolutions. The availability of a comprehensive software package also recommends this work. However, it also suffers from serious flaws, not least of which is the sloppiness of the manuscript. I don't feel a reviewer should serve as a copy editor, so I will only urge you to go over the figure numbering carefully and deal with many other minor issues such as identifying scale bars (Fig. 6).

We would again like to apologise for the error in Figure numbering and thank the reviewer for their feedback. This has now been rectified in the manuscript.

1. A major issue that wasn't addressed is the effect of probe diffusion and how it would limit the resolution of the method. A typical membrane diffusion coefficient for a small molecule like di4-ANEPPDHQ is greater than $1\mu\text{m}^2/\text{s}$. This means that it takes $\sim 1\text{s}$ for a dye molecule to sample a $\sim 1\mu\text{m}$ region or $\sim 0.01\text{s}$ for it to sample a 100nm radial region. But for SMLM, the molecules should be stationary on the time scale of the data collection. The authors claim "1825 \pm 688 localisations were collected per μm^2 (approximately one dye per 23 nm^2)". Unless the acquisition time (which was not mentioned) is faster than $\sim 0.000023\text{s}$, the physics should not allow identification of single molecules and the data should appear completely smeared out. Indeed, the data in Figures 5 and 6 appear to show continuous fluorescence (with a very sparse set of below-threshold dark pixels). The details on how the data were collected and single molecule detection was verified are completely missing.
 - a. This is an excellent point and is certainly something to consider with such PAINT approaches. In fact, diffusion of probes, when bound, has been exploited in the early implementations of PAINT techniques (Sharonov & Hochstrasser, 2006, PNAS). Firstly, we would like to clarify the localisations/micron² measurement is taken from the whole acquisition, and 23 nm^2 was a typographical error (now corrected to 584 nm^2 , lines:104-118). This number is derived from the number of localisations observed from the whole 10000 frames/micron². Thus, we get approximately 1 localisation every 5-6 frames/micron². This density is sufficiently low enough and separated in time to allow for the SMLM without the artefacts mentioned above with the integration time of 50 ms, and a complete acquisition time of 8 mins (lines:104-118). Similarly, the data in Figures 5 and 6 are from the whole acquisition, in essence a sum of the localisations, and show our density is sufficiently high enough to cover most of the membrane via MOM (showing few black pixels, pixel size 50 nm), and absolute requirement for this PAINT experiment. We have added text to further clarify this point in the main text and methods section regarding the data processing and collection and increased the detail in the section (lines: 104-118 and 326-377).

2. There was no discussion of the effect of dye bleaching or blinking. Were these common sources of problems in SMLM absent or were they corrected for?
 - a. Yes, this is also a salient point, and these are common artefacts associated with most fluorescent species in SMLM, and they are present here too. In our case, we correct for blinking (here defined in our analysis as an absence of signal with a one frame gap) using the grouping method, implemented in SMAP (Ries, 2020, Nat Methods). Bleaching is also present in our acquisition, and is a generally useful in PAINT acquisitions, i.e., more stable fluorophores often lead to longer and more complex PAINT experiments or do not allow PAINT. Here, the fluorophore generally bleached after 2 frames on average, and this behaviour is consistent with a lack of overlap between fluorophores in our density measurement (i.e., 1 localisation/5 frames/micron²) and allows us to perform this PAINT acquisition optimally. Our grouping and filtering parameters for processing in SMAP are outlined in the methods section (highlighted in the marked-up version, lines: 326-377).
3. Are the data in Figs 1-4 cartoons or simulations?
 - a. We understand that this could be a common misconception. For clarity we have now added some additional description in the figure title legends to make clear the type of data is contained in the Figures. For the reviewer, Figure 1 is a schematic of the PAINT approach and a real experimental example of a PAINT calculation for one localisation. Figures 2-4 are indeed simulations.
4. A review of existing alternate methods for nanoscale detection of rafts or lipid domains was missing.
 - a. We appreciate this feedback and have added further references around the topic of methods for nanoscale detection of lipid domains, in addition to those mentioned already which utilize environmentally sensitive dyes. This can now be found in the Introduction (lines: 63-69).
5. The experimental section is sparse. For example what are the details of the image acquisition; who supplies the ANEPPDHQ probe, etc.?
 - a. We thank the reviewer for this feedback, and we have increased the level of detail in the manuscript methods sections regarding reagent suppliers (highlighted throughout the methods section). For reference, we have highlighted in the marked version the sections already present in the methods relating to the image acquisition and processing (lines: 326-341 and 326-377).

Reviewer 2

The manuscript "Mapping membrane biophysical nano-environments" describes a method/software package to analyze membrane biophysical environment through changes of fluorescence emission of a solvatochromic probe (4-ANEPPDFQ) with so called "spectrally-resolved Single-Molecule Localisation Microscopy (SMLM)". By computing generalized polarization (GP) from the SMLM data, the author developed an algorithm that allows the detection of lipid domains (rather groups of similar GP spots) and validated the algorithm with both artificial membrane and live cell membrane. The cell membrane is of vital importance organelle to the cellular physiology and functions. Yet, there are limited tools to study the the dynamics and functional compositions and it is a hotly debated field. Ultimately, visualization

of so-called lipid rafts would yield significant understandings of such basic cellular functions. This tool not only extend the spatial information of SMLM data but also added extra dimension of information of the labelling environment and it would be highly useful in the biomedical fields.

The manuscript is well written in an easily understandable format with nicely done illustrations. However, numerous sections of the text refer to wrong figure numbers/panels and some were no existent (page 7) which is maybe due to some carelessness in manuscript revisions. Nevertheless, the methodology is sound, and I particularly appreciate the automated estimations of the 2 key parameters (radius and the deviance) which is often problematic in the success of image analysis field where key parameters require often subjective estimations.

We thank the reviewer for their positive comments and feedback, and we unreservedly apologise for the Figure numbering error, so we thank them for pushing through with the review.

For the manuscript, I have the following suggestions:

1. As a biologist, I cannot help but wonder if any experimental perturbation is applied to the cell and any changes were detected in the distributions of domain sizes, numbers. This is maybe beyond the scope of the paper. But expanding the biology of the method would excite many biologists in the field. Too often, new methodology was developed. But it remains in the expert labs and failed adopt widely due to the lack of example applications.
 - a. We thank the reviewer for this suggestion and agree his type of experiment would strengthen the manuscript. We have now included such data, where we have applied m-beta-cyclodextrin (MBCD) treatment to cells (to reduce cholesterol presence) and applied our PAINT approach. This new data can be seen in the new sub-panels of Figure 6 and Supplementary Figure 6. For the reviewer's convenience, we observed a decrease in the average GP value and thus membrane order (Figure 6c) consistent with the MBCD treatment effects and saw a relative shift of the deviation of the domains found in the data towards the disordered range (Figure 6d). Domain area distributions were found to be similar in the both the control and treated case (Supplementary Figure 6).
2. The method sounds solid with examples of successes in detecting clusters of domains with similar GPs. I am wondering if some discussions on the limitations of the techniques could be expanded. For example, how robust is the estimations of radius and deviance? This is particularly useful for the biologists where many lack necessary mathematics or statistics background to know the limitations of the methods. The model is built onto the assumptions that either clusters or non-clustered spots follow a normal distribution (which is likely true in most of the cases). But what if it does not (even in figure 5, panel e shows a not so normal distribution pattern)? Additionally, the method of paint-like SMLM yield relatively dense point SMLM data pattern. What is the minimum level of numbers of points needed for the algorithm to succeed? This is somewhat addressed in the discussion ("this may not be the most accurate summary statistic for data sets of particularly low density. ..."), but a bit more than a "low density" would be appreciated because "low" is not a scientific term.

- a. The need for discussion on the limitations of the methods are noted. We have now expanded this in the discussion and added some more detailed reasoning on the side of minimum density for analysis (highlighted, lines: 278-283).

Some minor (but really annoying!) issues:

1. Page 3: 1st paragraph “(here, membrane lipid order) as a point of interest (Figure 1a-c)..” It should be Figure 2?
 - a. Again we apologise for the Figure numbering error, and we have now corrected this.
2. Page 4: “Figure 1...” should be Figure 2?
 - a. Now corrected.
3. Page 7 “domains will not be identified(.) Further, the proportion...”
 - a. We have now corrected this typographical error.
4. Page 7 I am not sure where these figures refers to? “Figure 3g), i.e., if the mark values within spatially clustered points are similar to those outside of the clusters then domains will not be identified Further, the proportion of points assigned to domains (Figure 3h” But there is no panel 3h!
 - a. This is a Figure numbering issue again. Now corrected.
5. Page 7 “We first acquired data using established GUV preparations...” please define GUV.
 - a. We have now defined this in text. “giant unilamellar vesicle (GUV)” (highlighted, line: 205)
6. Page 8 Whether it is figure 5 or 4? “Examples of two MOMs from each data set are given in Figure 5a and 5b. The corresponding distributions of $P!$ values are given in Figure 4c and 4d and do not show statistically significant heterogeneity. This is expected as homogenous GPs are normally observed in these GUV types, with little difference in the hydration across the bilayers. The distributions of GP values acquired from each data type is depicted in Figure 4e. The average GP value of DOPC is shifted by roughly 0.4 units compared to DPPC:Cholesterol. Thus, in these model bilayer mixtures where spatial changes in hydration, and hence order, are not expected, the analyses do not detect domains.”
 - a. We have now amended the errors in Figure referencing and numbering.
7. Page 8: “the presence of excess di-4-ANEPPDHQ (80 nM) in DMEM (Figure 5a).” Figure 5A or 6A? Also, (80nM)...This is 4x higher concentration then the artificial membrane experiment and I am just wonder why?
 - a. For cell experiments we wanted to maintain high density coverage of localisations in the membrane as necessary for the GP-PAINT membrane mapping. We observed that this density could only be matched with a higher concentration for the cells. This is likely due to differences in the ability of the dye to reach the surface for a live cell compared to a nascent bare artificial membrane. In a cell it is likely that the presence of the extracellular matrix, and other components present within the membrane, may decrease the access for di-4-ANEPPDHQ. We have added some text to clarify this in the paper (highlighted, lines: 109-115 and 330-332)

8. Page 9, Figure 6? In this paragraph? "MOMs (Figure 5a-c). For each domain identified, the mean GP of the domain was compared to the global average GP of its ROI to determine the absolute difference in GP values across domains (Figure 5c). Results show two clear prominent peaks comprising clusters with GP values significantly above and below the global average, respectively. This suggests the existence of low order domains occupying regions of high order and the existence of high order domains occupying regions of low order. Additionally, the area of the convex hull of each cluster was determined (Figure 5d),"
- a. We have now amended the errors in Figure referencing and numbering.

Supplementary information

9. I could not install the PLAMA with the follow Error:
"Failed to install 'PLASMA' from GitHub:
Failed to install 'RSMLM' from Git:
Could not find tools necessary to compile a package
Call `pkgbuild::check_build_tools(debug = TRUE)` to diagnose the problem."
- a. The code has now been debugged and should now have a full description of the packages required for installation on the GitHub readme.
10. The supplied movie failed to display on my office desktop with windows media player. Although I was able to check it on a Mac OS computer, please check the compatibility of the movie. Here is some info of my office computer: Edition Windows 10 Enterprise Version 21H2 Installed on 10/21/2022 OS build 19044.3208 Experience Windows Feature Experience Pack 1000.19041.1000.0
- a. A new movie has now been saved in .mp4 format for viewing across platforms with some added experimental data.

Reviewer 3

In their manuscript "Mapping membrane biophysical nano-environments", Panconi et al. use single molecule localisation microscopy to investigate the presence of clusters with different lipid compositions within the plasma membrane. For this purpose, they employ the dye di-4-ANEPPDHQ, which exhibits different spectral properties depending on the surrounding lipid composition. According to the authors, the application of the dye from the outside leads to the integration of dye molecules into the membrane, creating a fluorescent point source for the use with SMLM algorithms to extract super resolution coordinates. The use of intensity data from two different color channels is used as an indicator for the type of lipid environment of the dye molecule (liquid disordered or ordered). This data is then used in the algorithms "P-check" and "JOSEPH" to detect the presence of clusters and their shape. The use of SMLM in combination with these algorithms allows to identify domains with higher order of lipid than would be possible with regular, diffraction-limited microscopy. The principles of the algorithms are first introduced and explored using simulations and later applied to vesicle preparations and living cells.

I very much like the concept of using di-4-ANEPPDHQ with SMLM. The ideas behind the authors' algorithms are clear and the implementation and documentation on their GitHub site is reasonably easy to follow. This technique harbours the potential to become an important tool for studying the structure of lipid membranes and the principle could be translated to other sensors. As of now, however, I feel that several of the underlying

assumptions need to be explored further before the authors' conclusions are sufficiently supported by the data. This is particularly necessary as the application of the authors' algorithms to cellular samples as shown in Fig. 6 is not fully convincing to me yet.

We thank the reviewer for the valuable feedback and apologise if the review was made difficult by the incorrect Figure numbering.

1. Are the chosen concentrations of di-4-ANEPPDHQ appropriate? My impression from the accompanying video is that the density of signals is rather high and it may be common to record overlapping emissions. It might be worthwhile to test several concentrations. Lower concentrations can also be helpful to better narrow down the number of photons from a single emission event (50-5000 photons sounds like a rather large range). Do the GP values change with concentration? There is a report (<https://pubs.acs.org/doi/10.1021/acs.jpcc.0c09496>) stating that this may be the case in cellular but not synthetic membranes.

a. This concentration is an important point when it comes not only to PAINT but more specifically to this implementation for mapping membrane domains. Critically the concentration of the dye or binder should be sufficiently high enough to give coverage of the whole system being observed within reasonable acquisition time (as here we deal with live cells). For probing the membrane this coverage demand is very high, and our goal was to use the concentration so that we have on average 1 localization on the order of the precision of the measurement (approximately the $\sim 584 \text{ nm}^2$ stated in the paper, i.e., approximately 1 localisation per $23 \times 23 \text{ nm}$ region), to allow such coverage. As identified, this can sometimes lead to overlapping PSFs, an issue with any PAINT acquisition, but increased by increasing concentrations like we use here. However, in our SMLM fitting and processing we filter out PSFs that arise from such cases (identified by widening of the PSF), thus, these situations should be minimally present within the final data. We have expanded the detail of our SMLM and data processing within the materials and methods (highlighted, lines: 326-377)

2) The contribution of background noise must be determined for each color channel. This should be done in the absence of sample and/or dye. In particular, for the cellular samples shown in Fig. 6, this must also be done with cells but without dye as several endogeneous molecules may create endogeneous signals.

a. We agree this is an important to determine. We have now included data on these requested controls. For glass only we see binding to the slip (Supplementary Figure 1), present in all the data, however, as seen from the GUV data often these are of too low intensity to survive our processing pipeline (hence clearly defined regions of membranes in our MOM representations in the full FOV, Figure 5). With cells only (Supplementary Movie 1, Panels A and B) we see an expected dramatic decrease in any events observed. Notably, we do see punctate autofluorescent signals in cells, appearing like vesicles, but these again are filtered out due to a lack of matching signal in the far-red (and are generally bleached completely by 5000 frames), unlike specific di-4-ANEPPDHQ signals.

3) How exactly do you deal with chromatic aberration (CA) and sample drift? In particular, accurate correction of CA is crucial to calculating the GP value for an individual dye emission.

- a. We agree this is a crucial consideration for multi-channel/multi-wavelength imaging as seen in the manuscript. This is first dealt with by registration of the detected localizations to align the data acquired at different wavelengths and provide a transformation accounting for translation, rotational and scaling difference between the channels. By aligning the images and localisations therein, SMAP compensates for the spatial shifts caused by chromatic aberration, ensuring accurate localization data across channels. We then run a second localization fitting process incorporating the transformation calculated in the previous step, where the centre of each PSF in the reference channel (here, the short wavelength channel) is determined one by one, and then a fitting is done to find the correct and corresponding PSF in the target channel (the long wavelength channel). Lack of a PSF in the target channel or poor-quality fitting leads to exclusion of that localisation from the data. Thus, poor correction would lead to very few localisations in the final data set. We have given a more detailed breakdown of our SMAP processing pipeline in the methods section (highlighted, lines 326-377).

4) Connected to the above: What is the true resolution of your setup? While SMAP may be able to determine each individual localization with 0-50 nm precision, after CA, drift correction, and grouping, the uncertainty propagates and may be much higher than assumed. Is the 50 nm search radius really appropriate for the resolution of the data?

- a. The reviewer rightly points out that we should be mindful of errors that can reduce the effectiveness of the measurement, e.g., poor localization precisions etc. This will be associated with the point patterns; however, the search radius is more related to the density of points, i.e., this needs to be large enough to capture neighbouring points, but small enough so that we do not average out/lose nanoscale changes in the data. Here, we estimate the radius from the peak of the Ripley's K function for the data, and this is generally a good approximation for the average radius of the clusters within the data (this was originally outlined in the supplementary text and methods section and have directed the readers more clearly to there. Line: 392). Given our average localisation density is targeted to give us on average one localisation per 584 nm², i.e., on localization per ~25x25 nm region separated in time. This radius, therefore, should be minimally sufficient to capture neighbours meanwhile making us sensitive to changes on the scale of the data.

5) Is the GP value only dependent on the surrounding lipids? What about the presence of (trans-)membrane proteins?

- a. It can be seen in the raw data that the dye is mobile in the membrane, albeit fluorescent for a short period. Thus, this should allow the di-4-ANEPPDHQ to explore a variety of environments, and the movement is likely permitted by being in the more lipophilic regions of the membrane. It has been demonstrated previously that di-4-ANEPPDHQ is not affected by the presence of membrane inserted peptides derived from viral and transmembrane proteins (Jelena Dinic, Henrik Biverståhl, Lena Måler, Ingela Parmryd, Laurdan and di-4-ANEPPDHQ do not respond to membrane-inserted peptides and are good probes for lipid packing, *Biochimica et Biophysica Acta (BBA) - Biomembranes*, Volume 1808, Issue 1, 2011). We agree this is an important point and we have now added this detail to the main text (highlighted, lines: 100-102)

6) May the GP value be affected by the orientation of the membrane/dye relative to the

incident light? You assume a perfectly flat surface in your simulations but real samples may be unevenly attached. For example, in Fig. 5b, it seems as if "low GP spots" are associated with black spots (without signal, perhaps because they are not in the plane of excitation?).

a. This is a good point regarding dye orientation. The reviewer rightly points out that we have dark spots in the data, which are due to low coverage, i.e., no localisations within the pixel bin, and could arise from dyes not being optimally in the plane of excitation for live cells. However, even in this case it's unlikely the GP value will be affected for the dye, i.e., the ratio of the short and long channel, but more the overall intensity of emission. This means that in area where we may have sub-optimal excitation and at the extremes of the GP scale, we may filter these events due to our stringent filtering to preserve the quality of the localisations used for analysis, i.e., rejecting localisations with very low photon numbers, e.g., like those seen on the slip or cells only.

7) I presume that the localisations in your simulations are stationary but this may not be the case in cellular samples. Can you estimate if lateral drift of lipids (and perhaps even full domains) occurs during your recordings? Perhaps it is possible to split the recordings into multiple sub-stacks and check if the features are stationary over time or not?

a. We agree with the reviewer that its certain that the lipids will be moving in all settings (artificial and live cell membranes), thus domains themselves may move. As suggested, we have now included subsampled example of the MOM allowing the potential movement domains to be observed (Supplementary Figure 5). Given this is it still difficult to make these estimates, as the diffusion of lipids is still at a much quicker scale than the sub-setted data (approximately 2 min windows bound by the need for a minimal density of localizations as described previously to perform the analysis).

8) Is MOM really required and can the same result not simply be achieved by rasterizing the original GP point pattern (i.e. binning Fig. 3a)? Also, you state that MOM is explained in "Methods and Materials", which it is currently not.

a. Firstly, we apologise for the omission of MOM method section, and it is now within the manuscript (highlighted, lines: 437-451). This approach of averaging the GP value could certainly be taken for the MOM, and tests in the lab show that the mean value for the GP in the bin seems to be very close but not equivalent to the photon histogram approach. We prefer to use the approach utilizing the photon counts from the data, as this more closely aligns to the traditional methods for calculating the GP from two conventional diffraction limited images (Owen DM, Rentero C, Magenau A, Abu-Siniyeh A, Gaus K. Quantitative imaging of membrane lipid order in cells and organisms. Nat Protoc. 2011 Dec 8;7(1):24-35.).

9) It seems to me that the underlying assumption of the manuscript is that there is a clear cutoff between high and low order domains. Is this assumption valid? The domains in Fig. 6 appear much less clearly defined. Do you find differences within the domains regarding GP value "density"? If present, maybe these could be analysed as well?

a. The reviewer is correct that the analysis in the paper is predicated to find domains, thus there is an assumption that there is a "cut-off". We identify domains as such where we have a clustering of points within the data which deviate significantly from the mean GP for that region. In the case where we shouldn't have this separation, i.e., the artificial

membranes we do not see this deviation and no domains are detected. Whilst when we move to the cell experiments, we do now detect domains.

10) It would be a good control for the applicability to real samples, if you could demonstrate (dynamic) changes to the lipid composition of your model system. Perhaps depletion of cholesterol in your cellular samples using methyl- β -cyclodextrin could be performed?

- a. We agree, and we have done as suggested. We have now used methyl-beta-cyclodextrin (MBCD) to deplete cholesterol. New data is included in Figure 6 and Supplementary Figure 6. We were able to detect a significant shift in the average GP value per ROI towards the disordered end of the scale (i.e., more negative). Further, we show the proportion of domains that deviate from the mean GP toward the disordered end are enriched when we use MBCD when compared to the control.

Dr. Gabriel Stölting
Center of Functional Genomics
Berlin Institute of Health at Charité - Universitätsmedizin Berlin

To the Reviewers,

Firstly, we would like to thank you again for your valuable feedback, which have allowed us to once again improve the manuscript.

We respond to each of your points below (text in red) and provide a new marked version of the manuscript and Supplementary Information, with changes highlighted for your ease.

REVIEWER COMMENTS

Reviewer #1 (Remarks to the Author):

I appreciate the authors' effort to make the paper clearer and eliminate all the original errors in the text and figures. Now that I can understand the paper better, I still have some concerns and suggestions.

1. The authors did not address my major concern adequately. They now correct the typo on the single molecule localization, but it is still small, at 23nm, on the scale of diffusion within the time per frame of 50ms. These small organic molecules would diffuse out of the 23nm region in less than 1ms, if my estimate of a $\sim 1\mu\text{m}^2/\text{s}$ diffusion constant is close to correct. How can such a fleeting molecule be localized with such precision within 50ms? Do the authors have some reason to believe that my estimate of the diffusion is way too high? It would be interesting to do a simple FRAP experiment on di-4-ANEPPDHQ labeled cells or GUVs to see what D is. Perhaps the localizations represent a sub-population of dye molecules that are forming less mobile aggregates in the membrane; such aggregates could have altered spectral properties and compromise the interpretation of the data in terms of lipid order. Incidentally, the original work by Sharonov & Hochstrasser, 2006, used immobile lipid vesicles as $<100\text{nm}$ targets for Nile red; so the diffusion of the dye within these spatially restricted membranes was not a concern.

Firstly, we thank the reviewer again for giving us a chance to clarify our PAINT based approach, and in general the assumptions of these approaches. We feel it is important to address what could be a fundamental misconception here about PAINT, both regarding the diffusion of the dye and what we report as our final density of localizations.

To start, we think the reviewer's estimate of the diffusion coefficient is reasonable, and it has been measured relatively recently in cell membranes by RICS, and is close to what the reviewer suggests, with a mean of $\sim 0.8\mu\text{m}^2/\text{s}$ (Gardeta SR, García-Cuesta EM, D'Agostino G, Soler Palacios B, Quijada-Freire A, Lucas P, Bernardino de la Serna J, Gonzalez-Riano C, Barbas C, Rodríguez-Frade JM, Mellado M. Sphingomyelin Depletion Inhibits CXCR4 Dynamics and CXCL12-Mediated Directed Cell Migration in Human T Cells. *Front Immunol.* 2022 Jul 12;13:925559. doi: 10.3389/fimmu.2022.925559). It must be noted in this case, that RICS is a technique which is sensitive to only the diffusing molecules in the region of interest, so this is likely an overestimate due to the moving average applied to data to remove relatively immobile fractions. Therefore, it's likely the average diffusion coefficient from all dye molecules in the sample/membrane could be lower.

Regarding the average density of the data (1 localization per 23x23 nm region, 584 nm²), this is the final average density of all the localizations. To be clear this is **not the precision of our localization or the area of our observation**. As is the standard for PAINT, to achieve substantially high coverage of the sample the pool of excess binders/probes in solution allows constant re-probing of the same environments even for diffusing species. This is demonstrated thoroughly in the seminal uPAINT paper (Giannone G, Hosy E, Levet F, Constals A, Schulze K, Sobolevsky AI, Rosconi MP, Gouaux E, Tampé R, Choquet D, Cognet L. Dynamic superresolution imaging of endogenous proteins on living cells at ultra-high density. *Biophys J*. 2010 Aug 9;99(4):1303-10. doi: 10.1016/j.bpj.2010.06.005). For example, in the uPAINT work they almost exclusively work with diffusing species, with a wide range of diffusion coefficients, i.e., a GPI-anchored GFP molecule which approaches the diffusion speeds estimated for di-4-ANEPPDHQ stated above. Here, they can localize the GPI-GFP not only once, but several times in consecutive frames, before bleaching, all improving the final localization density of the data. As the reviewer points out, diffusion of the binder is an inherent property of most PAINT and uPAINT techniques and is referenced heavily in the papers discussed. In uPAINT, this was in fact leveraged, if you can detect your molecules in subsequent frames, one gets increased overall localization density in the data from a single binder exploring different regions. Again, this is demonstrated in their work with the GPI-GFP where they get 4-5 subsequent localizations from the trajectory of a single GPI-GFP. Even more importantly the integration time for a single frame in their experiments is 50 ms, and even more compellingly the GPI-GFP is measured to diffuse at speeds like those we have discussed above. Therefore, the notion of diffusing molecules during frame integration seems to not be an inherent issue in PAINT, and is fact an advantage, if your probe is stable enough to be observed in several frames whilst diffusing. In our case, however, with di-4-ANNEPDHQ we did not observe such stability, with our detected events lasting only 1-2 frames (likely due to stringent filtering on our side for the localization).

Coincidentally, our localisation precision is on the order of 20-30 nm per event. This is relatively standard for single particle tracking (SPT) experiments of membrane probes which are designed to measure diffusion coefficients. Detected PSFs that would result from a “blurring/smearing” due to the diffusion over scales larger than the PSF of ~300 nm (**not** the 23 nm final density or the 20-30 localisation precision), would be not be detected with the conventional PSF fitting due to their much-reduced SNR (i.e., the brightness of the probe in that time window is spread over more pixels).

2. The time for an experiment is about 8 minutes, if I understand correctly (this should be made clearer to readers right in the Figure legends for Figs. 5 and 6). This could be a real limitation for any practical application of the method as small lipid domains are likely to have much faster turnovers.

We agree and have now referred to the length of the experiments more clearly within the text, methods, legends (as requested), and new supplementary figures, whilst also adding this to the discussion. In relation, to the reviewers point regarding turnover of lipid domains, we have now added two supplementary figures (Supplementary Figures 5 and 6) that address the length of time and movement speed of domains, and their effects on measurements via JOSEPH. For this case, of lipid domain turnover, we observe that the domains need to be present for approximately 25-30 % of the

acquisition to remain detectable in the final data taken as a whole (on the order of 100-150 s, Supplementary Figure 5), thus the reviewer is correct, we are unlikely to detect transient domains if they appear for a time shorter than this with the originally proposed approaches (i.e., 2000 frame blocks, or taking the data as a whole at the end), however, this will be context specific. Importantly, this data was used as a demonstration for the detection algorithms, and in other non-membrane data, domains may be much more stable.

With this in mind, and taking lead from an excellent suggestion of another reviewer, we have applied a sliding window approach to our live cell data to try to address this shortcoming. As suggested, we have used a window width of 2000 frames (to maintain good point density), but this time moved the window by 10 frames per block (equal to 500 ms; Supplementary Movie 2). This shows in our live cell data that we can detect nanoscale domains, and assess the stability, by scoring the similarity of the current and following time windows (Supplementary Figure 7). We observed in our data that the similarity in structures identified declines to > 0.1 within 21.5 s (Supplementary Figure 7 b). This suggests a significant reduction in spatial overlap of domains within the width of one window. This suggests that dynamics of domains in our data occur on average at timescales less than 2000 frames (100 s). We have suggested future avenues and methodologies in the discussion that could potentially access shorter dynamics in the context of membrane data (lines: 305-310, 319-324).

3. It would be valuable to add something to the discussion about what specific cell biological questions might be addressed with this methodology.

Agreed. We have now briefly made some suggestions within the discussion on application of this methodology, and its potential in conjunction with other new probes and methods, to other investigate cell-based questions (lines: 305-310, 319-324).

Reviewer #2 (Remarks to the Author):

The authors addressed all my concerns and I think adding the m-beta-Cyclodextrin treatment experiments strengthened the methodology. I have no objection for the publications as it is.

We thank the reviewer for their recommendations and agree implementation of their suggestions significantly improved the manuscript.

Minor issues:

Figure 1 (should be 2!) (page 4) was still not labelled correctly in the merged document which is the one I read.

We have checked the manuscript and find no error in the Figure numbering. This was the correct Figure reference on page 4 of the previous version, as intended and corrected last time.

The merged file contains 2 copies of the main manuscript. The authors (did all of them?!) should read it carefully.

The merged document is generated by the mts site, and likely has merged two

copies of our manuscript, we did not knowingly upload the manuscript in a way it would produce the duplication.

Reviewer #2 (Remarks on code availability):

NA

Reviewer #3 (Remarks to the Author):

I would like to thank the authors for considering and responding to my comments. Overall, the manuscript has greatly improved and almost all of my issues have been satisfactorily addressed. However, in my view, there are two remaining problems with the answers to my previous comments #7 and #6 that require further consideration before the author's claims are sufficiently supported. I pasted my original comments and the author's replies below and put my novel comments underneath.

My original comment #7 and the author's reply was as follows:

"7) I presume that the localisations in your simulations are stationary but this may not be the case in cellular samples. Can you estimate if lateral drift of lipids (and perhaps even full domains) occurs during your recordings? Perhaps it is possible to split the recordings into multiple sub-stacks and check if the features are stationary over time or not?

a. We agree with the reviewer that its certain that the lipids will be moving in all settings (artificial and live cell membranes), thus domains themselves may move. As suggested, we have now included subsampled example of the MOM allowing the potential movement domains to be observed (Supplementary Figure 5). Given this is it still difficult to make these estimates, as the diffusion of lipids is still at a much quicker scale than the sub-setted data (approximately 2 min windows bound by the need for a minimal density of localizations as described previously to perform the analysis)."

I would like to thank the authors for analyzing the sub-setted data as suggested. While I am certain that individual lipids will be moving rapidly, I don't necessarily assume the same for the lipid domains which are the object of investigation in this manuscript. There are likely several theories on the organization of these domains but I find it conceivable that they are (at least partially) tethered to membrane proteins and thus much less mobile than individual lipids.

Unfortunately, I find the results worrisome. It is hard to say from the representative images in Sup. Fig. 5 alone but it appears that the domains are not stable over the typical duration of the recordings used in the manuscript (8,000-20,000 frames accordings to Materials & Methods). This is also what I take from the author's answer to my comment. Should the domains even move significantly within the stated minimum window of 2 minutes, it would be even more problematic.

If these limitations were true and the temporal resolution of the recording system is

much below the movement speed of the observed lipid domains, the resulting maps are mere averages with a lower resolution than the stated “nanometer(s)”. Using short and standardized recording protocols will clearly minimize this error yet the actual resolution will be unknown and may be different for each sample.

I strongly feel that this issue needs to be investigated in more detail beyond the representative example in Sup. Fig. 5. I'm not sure if it would be worth the effort but simulations that include movement of the lipid domains may help to determine limits for the algorithms. Such simulations may also help to further optimize the minimum number of frames required. Also, it would be helpful to determine the speed of movement of the lipid domains. If feasible, one could use a moving analysis window (of 2 minutes or lower, if possible) and define the clusters using JOSEPH. This may allow to visualize and quantify the behavior of these domains over time.

We thank the reviewer for their further excellent comments regarding this point on movement of domains. We have sought to investigate the effects on the domain determination of both the transient appearance and dissolution of domains (akin the diffusion of lipids) and the movement of whole domains, as suggested. We present these simulated data here in our response (Supplementary Figure 5 and 6).

For diffusion of lipids and dissolution of domains, we show that we are still sensitive to regions of differing membrane order when the domains are present and stable for as low as 25% (equivalent to 100 s) of the whole 8-minute acquisition, and determination of the domain shape is improved the longer the domain is intact.

For lateral drift of the domains, we have simulated the movement of an intact domain within the region of interest at two different speeds (~45 nm per 100 s time window, and ~200 nm per 100 s time window). We see, as expected, that when the drift/movement of domains is small, and we can recapitulate the shape well. When the drift is high (800 nm over the whole acquisition) we are still able to detect the regions of higher order, but the underlying ground truth is met less well. Importantly, we are still sensitive to the membrane order in these sub-diffraction regions when taking the data (i.e., the full 8 mins).

We acknowledge this can be a limitation of the method as presented previously in this context, i.e., measuring domains on live cells which may be highly dynamic. Therefore, as suggested by the reviewer we have implemented the sliding window approach on our data, now shown in the redrafted Supplementary Information (Supplementary Figure 7). Briefly, we implemented a sliding window starting from the first frame (and 2000 frames in width = 100 s) and moved this window by 10 frames (0.5 s), repeating the JOSEPH analysis and then calculate the IoU between the initial window (frames 0-2000) and all subsequent windows. We observe the IoU drops to half its peak value within a 100-frame delay (equivalent to 5 s) and below 0.1 at a frame delay of 430 (equal to 21.5 s, akin to complete lack of similarity). This suggests a significant reduction in spatial overlap of domains within the width of one window and thus the dynamics of the domains detected in our data occur at timescales less than 2000 frames (100 s). We would like to state that this data was chosen initially demonstration for the detection algorithms, and in other non-membrane data, domains may be much more stable and less dynamic.

However, we have discussed these limitations in the context of this membrane data and made suggestions within the discussion on application of this methodology, and its potential, in conjunction with other new probes and methods to study these dynamics at shorter timescales (lines: 305-310, 319-324).

My original comment #6 and the author's reply was as follows:

"6) May the GP value be affected by the orientation of the membrane/dye relative to the incident light? You assume a perfectly flat surface in your simulations, but real samples may be unevenly attached. For example, in Fig. 5b, it seems as if "low GP spots" are associated with black spots (without signal, perhaps because they are not in the plane of excitation?).

a. This is a good point regarding dye orientation. The reviewer rightly points out that we have dark spots in the data, which are due to low coverage, i.e., no localisations within the pixel bin, and could arise from dyes not being optimally in the plane of excitation for live cells.

However, even in this case it's unlikely the GP value will be affected for the dye, i.e., the ratio of the short and long channel, but more the overall intensity of emission.

This means that in area where we may have sub-optimal excitation and at the extremes of the GP scale, we may filter these events due to our stringent filtering to preserve the quality of the localisations used for analysis, i.e., rejecting localisations with very low photon numbers, e.g., like those seen on the slip or cells only."

I generally agree with the authors that the filtering and ratioing does remove a lot of low-quality localizations. The improved description of the settings used for filtering is highly appreciated. However, I do have remaining doubts as to the issue of a non-flat membrane surface. As the authors are aware, the inherent axial chromatic aberration of the objective results in a slightly different focal plane for each color. Shifting the position of a fluorophore in the z-axis may thus differentially affect the signal from the two color channels. I presume that this affects all experimental recordings to some degree as the membranes (synthetic or cellular) will rarely be perfectly flat and contributes to the overall noise of the system. However, I remain worried that this results in a systematic error in some areas such as near the border of cells or around larger deviations from a flat membrane (due to artifacts but perhaps also vesicular structures). If this were the case, this might be detected by having systematically different PSF sizes near these structures.

We agree with the reviewer that this certainly could play a role in the regions of the cells suggested (i.e., cell edges). We have updated the methods section further to make clear where we select our regions for analysis from. We were also concerned by the cell border, thus, only take regions for analysis from within the cell boundary, therefore, limiting the effect of this structure on the data (now highlighted in the methods). Further the cell type used in these experiments is a fibroblast line that has been previously utilised in other single molecule studies of the apical membrane due to its relative flatness during "longer" acquisitions, e.g., photothermal single particle tracking experiments, which can last from several hours up to days (Duchesne L, Oceau V, Bearon RN, Beckett A, Prior IA, et al. (2012) Transport of Fibroblast Growth Factor 2 in the Pericellular Matrix Is Controlled by the Spatial Distribution of Its Binding

Sites in Heparan Sulfate. PLOS Biology 10(7): e1001361). Further, fibroblasts have been the cell type of choice to minimize such artefacts for other PAINT based approaches (Giannone G, Hosy E, Levet F, Constals A, Schulze K, Sobolevsky AI, Rosconi MP, Gouaux E, Tampé R, Choquet D, Cognet L. Dynamic superresolution imaging of endogenous proteins on living cells at ultra-high density. Biophys J. 2010 Aug 9;99(4):1303-10. doi: 10.1016/j.bpj.2010.06.005.).

Whilst these cells are relatively “flat” we acknowledge and agree with the reviewer that there are still likely to be small undulations and protrusions formed as a natural process of the living cell. The size of these undulations/protrusions on the apical surface are normally small in fibroblasts (sub-micron scale), thus, we would not expect large changes in the spectroscopic properties of the dyes under widefield detection on a glass surface between the flatter membrane and membrane protrusions. Similarly, in the case of vesicles, whilst the dye is labelling the sample for short lengths of time before imaging, and bleached within 1-2 frames of binding during, we could get some vesicles that are labelled during the acquisition diffusing and increasing the imaging depth. In this case one would expect to see much more heterogeneity in our live cell data when compared with our GUV experiments, if this issue arose commonly. These phenomena are, however, something that could be explored in more detail with new imaging modalities to examine such structures (we have now also added this point to the discussion, lines; 305-310). This could lead however to artificial enrichment of point density in certain regions of the data; thus, we have re-assessed the data with this in mind. We tested each of our ROIs for its similarity to a completely spatially random distribution of points, i.e., if there were regions like undulations/protrusions in the data that did artificially and non-randomly increase the point density, they would be detected. One would expect from this PAINT acquisition that the points themselves (not the GP values) would be randomly distributed. We found 43 out of 49 ROIs passed this test, thus, the effects described are likely limited, but to err on the side of caution, we now have taken only these ROIs forward for analysis.

Whilst we have attempted to attenuate contributions from axial chromatic aberrations, we agree with the reviewer they can be an inherent problem of microscopy systems. Here, this study used a Nikon CFI Apochromat TIRF 100X objective with chromatic corrections in place, limiting the aberrations. Upon investigation, we have detected slight variations in PSF width between the two channels. However, we could not find any indication that the detected total photon counts per emission event and hence the GP-values were affected by axial chromatic aberrations. We have validated that the system is capable of robustly determining the photon count in z-stacks of bead samples for 300 nm around the focal plane. While the PSF width is measurably affected by both the wavelength and the axial position, the fitting algorithm is capable of robustly determining the total photon count for out-of-focus PSFs. Here, the fitting algorithm in SMAP can recover the constant total photon count, as the photons are merely distributed over a larger area on the camera sensor for out-of-focus PSFs. We therefore believe that axial chromatic aberrations have not affected the results of this study in any meaningful way.

Dr. Gabriel Stölting
Center of Functional Genomics
Berlin Institute of Health at Charité - Universitätsmedizin Berlin

Reviewer #3 (Remarks on code availability):

The code works on my machine and I can analyse the provided data sets. I have not attempted a full reproduction of the data presented in the manuscript, however. The repository appears to be reasonably well structured but could profit from more documentation, including more commenting within the code.

REVIEWERS' COMMENTS

Reviewer #1 (Remarks to the Author):

The authors have provided a clarification in their response to my main concern. I thought they were claiming 23nm resolution, but they are actually reporting 1 localization per 23x23 nm region. I now understand that they are referring to the density of localizations rather than the resolution. While having a high density of localizations is a prerequisites for attaining good domain resolution, it is by far not the only factor. The authors claim nano-scale resolution, but only demonstrate this in their simulations. They need to demonstrate this with experimental data, containing real measurement noise and probe diffusion. They keep referring to their method as detecting "nano-environments", but have not established the resolution of their measurements.

We thank the reviewer for their comment. The live cell data will have all the noise and probe diffusion described above here and addressed in our previous response. We have now included a sentence in the discussion acknowledging we have not directly established this resolution in the live data (line 314). We hope this give thorough insight into the current limitations of the approach.

I am satisfied that they now more clearly state the slow acquisition time required for the measurements.

I am satisfied that they have added some potential biophysical questions that might be addressed with the method.

We thank the reviewer again for their comments.

Reviewer #3 (Remarks to the Author):

The authors have significantly improved the manuscript and the potential but also the limitations of the method are now much clearer. I would be very happy to see this paper being published in Nature communications and I'm eagerly looking forward to see how the authors and other groups will employ the developed algorithms in the future.

Dr. Gabriel Stölting
Center of Functional Genomics
Berlin Institute of Health at Charité - Universitätsmedizin Berlin

We thank the reviewer again for comments throughout the process, which have significantly improved the manuscript.